# Distributionally Robust Feature Selection

**Maitreyi Swaroop**
Machine Learning Department
Carnegie Mellon University
mswaroop@andrew.cmu.edu

**Tamar Krishnamurti**
Division of General Internal Medicine
University of Pittsburgh
tamark@pitt.edu

**Bryan Wilder**
Machine Learning Department
Carnegie Mellon University
bwilder@andrew.cmu.edu

## Abstract

We study the problem of selecting limited features to observe such that models trained on them can perform well simultaneously across multiple subpopulations. This problem has applications in settings where collecting each feature is costly, e.g. requiring adding survey questions or physical sensors, and we must be able to use the selected features to create high-quality downstream models for different populations. Our method frames the problem as a continuous relaxation of traditional variable selection using a noising mechanism, without requiring backpropagation through model training processes. By optimizing over the variance of a Bayes-optimal predictor, we develop a model-agnostic framework that balances overall performance of downstream prediction across populations. We validate our approach through experiments on both synthetic datasets and real-world data. [1]

## 1 Introduction

Many real-world applications impose significant constraints on data collection for machine learning models. These constraints often stem from factors such as limited budgets, privacy concerns, or the operational costs associated with acquiring each data point. For instance, in healthcare, a hospital system aiming to implement a predictive screener across diverse patient populations must often contend with limitations on the number of questions permissible due to patient burden, time constraints, and regulatory considerations. In such scenarios, the ability to identify a minimal yet highly informative set of features—*feature selection*—becomes paramount. Oftentimes, we might collect pilot data on a large number of features in order to make the operational decision of which to collect in deployment. For example, medical screeners are often developed with a larger number of questions that are then cut down to a smaller number for general-population usage; e.g., the PHQ [Spitzer et al., 1999] is a 26-item mental health survey that is typically reduced to standard 9 or 2-question versions in practice [Kroenke et al., 2001, 2003].

The challenge intensifies when the selected features must ensure reliable model performance not just on average, but across varied and potentially shifting underlying data distributions. This necessitates a *distributionally robust* approach to feature selection, ensuring that no particular subpopulation is inadvertently neglected by models trained on the selected features. Furthermore, in many practical settings, the specific downstream inference model that will ultimately use the collected data may not be known at the time of feature selection, or multiple different models might be employed. Therefore,

---

[1]Code for implementing our method is available here (linked).

39th Conference on Neural Information Processing Systems (NeurIPS 2025).

an ideal feature selection methodology should be *model-agnostic*, providing a core set of features that are broadly useful without being tied to the idiosyncrasies of a particular predictive algorithm.

This paper addresses the problem of selecting a limited subset of features from a larger pool, using historical data, to facilitate accurate and robust predictions across diverse populations, all while adhering to a collection budget. We aim to identify features that are robust to distributional shifts, ensuring that the model performs well simultaneously across a variety of specified subpopulations (e.g., institutional settings, clinics, or demographic groups).

To our knowledge, this problem has not been previously studied in the literature: feature selection methods focus on a single distribution of interest, while distributionally robust optimization (DRO) methods attempt to find a single model that performs well on all populations, instead of selecting a limited number of features that can then be used to train high-performing models for each downstream population. The intersection of the two problems complicates both. On the one hand, we require an approach to feature selection that is model-agnostic and naturally extends across multiple populations. On the other hand, standard DRO methods do not apply because we have a discrete optimization problem of selecting features instead of directly optimizing over a single model. Our approach formulates this task by introducing a continuous relaxation of the discrete feature selection problem which injects synthetic noise into the observation of each covariate. We then develop an analytical simplification of the relaxation which eventually allows us to solve it with standard stochastic gradient descent-style methods.

Our main contributions are as follows:

1. We propose a novel, model-agnostic method for distributionally robust feature selection. This method identifies a subset of features that minimizes the maximum expected loss (or error) across potential data distributions, without requiring assumptions about the specific architecture or differentiability of the downstream predictive model.

2. We reframe the combinatorial problem of feature selection into a continuous optimization framework. This is achieved by introducing parameters that govern a noise injection process, effectively creating a differentiable measure of each feature's utility and permitting tractable, gradient-based optimization for selection.

3. We demonstrate the efficacy of our approach through experiments on both synthetic and real-world datasets, comparing its performance against naive selection strategies and existing distributionally robust optimization (DRO) baselines adapted for feature selection.

## 1.1 Related work

Our work lies in the intersection of distributionally robust optimization and feature selection. While there is a rich body of work that focuses on each topic individually, to our knowledge no previous work studies the intersection – selecting a set of *features* that will allow a high-performing model to be trained for each subpopulation, as opposed to training a single model that works well everywhere.

**Feature selection:** Classical feature selection methods primarily fall into two categories: combinatorial optimization approaches [Guyon and Elisseeff, 2003, Kohavi and John, 1997], and embedded methods [Breiman, 2001, Tibshirani, 1996]. In the combinatorial optimization vein, a long line of work develops greedy strategies or forward/backward selection heuristics for variable selection, historically focused on linear models. Even for linear models, selecting the best set of at most $k$ features is known to be NP-hard [Natarajan, 1995]. Das and Kempe [2011], provided theoretical approximation guarantees for greedy selection in linear models via a connection to submodularity. Khanna et al. [2017] provide faster algorithms for the same problem. However, there is no direct path to extend these methods beyond linear models, or to incorporate robustness across multiple distributions as a goal. Lasso regression [Tibshirani, 1996] is a classical method frequently used for feature selection, which leverages an $\ell_1$ penalty to induce sparsity. However, the Lasso formulation does not account for robustness to distributional shifts and again bakes in an assumption of linearity. Zhao and Yu [2006] highlight potential pitfalls of Lasso-based feature selection, showing that it may not consistently select the correct variables under certain correlation structures. Recent advances in embedded methods have explored heterogeneous feature selection, though these address different problem settings from our work. Yang et al. [2022] propose locally sparse neural networks (LSPIN) that learn sample-specific feature subsets, allowing different features for different samples. Similarly, Svirsky and Lindenbaum

[2024] introduce interpretable deep clustering (IDC) that performs cluster-level feature selection. While these methods advance feature selection for heterogeneous populations, they fundamentally differ from our approach: our method selects a single global feature subset that must perform well universally across all known population groups, a constraint motivated by practical deployment scenarios where systems require a fixed set of features for all users. Previous work has also proposed the use of random noise injection for feature selection in single population settings [Grandvalet, 2000], as well as through Bayesian relevance estimation methods [Neal, 1996, Tipping, 2001]. However, tunable-noise-based variable selection has seen limited adoption since it was originally proposed, in contrast to the large body of work on noise as a form of regularization [Bishop, 1995]. In this work, we revisit noise-based relaxations in the distributionally robust setting and show how the optimization problem can be reformulated in ways that create significant, previously-unrecognized advantages. Crucially, our approach separates variable selection from predictive model fitting, making it agnostic to the downstream model and eliminating the need to differentiate through model training – which may be infeasible for frequently used models like decision trees or random forests.

**Distributionally Robust Optimization (DRO):**   DRO provides a principled framework to account for worst-case model performance under distribution shifts, contrasting with traditional ML methods that optimize only for average performance. Duchi and Namkoong [2020] formalize this approach by providing finite-sample minimax bounds for uniform model performance across test distributions. Group DRO, which focuses on worst-case performance across predefined population groups, is particularly relevant to our work. Sagawa et al. [2019] introduce a group DRO approach for neural networks that explicitly optimizes for the worst-case loss over known population groups, demonstrating improved performance on underrepresented groups compared to standard empirical risk minimization. Similarly, Hashimoto et al. [2018] demonstrate that minimizing the worst-case risk across groups can prevent representation disparity in sequential learning settings.

## 2   Problem formulation

We assume our input $(X, Y) \sim P$ to be covariates $X \in \mathbb{R}^m$ and outcomes $Y$. We wish to select a subset of $k < m$ variables from $X$ that are most informative for predicting $Y$, while being robust to shifts in the underlying distribution $P$. Let $\mathcal{I}$ denote a set of indices corresponding to the selected variables, with $|\mathcal{I}| = k$. The selected sub-vector is $X_{\mathcal{I}} \in \mathbb{R}^{|\mathcal{I}|}$. Informally, we wish to solve the problem:

$$\min_{|\mathcal{I}|=k} \max_{P_i \in \mathcal{P}} \mathbb{E}[\text{Loss of a model trained on } (\tilde{X}_{\mathcal{I}}, Y) \sim P_i]$$

where $\mathcal{P}$ represents the set of distributions we wish to ensure robust performance on. The above can be equivalently expressed as the problem of choosing a binary mask $\boldsymbol{\alpha} \in \{0, 1\}^m$, $\sum_i \alpha_i = k$ where the model observes $\tilde{X} = \boldsymbol{\alpha} \odot X$. To formalize the problem, let $\mathcal{L}$ be a loss function (we will work with the mean squared error throughout), and

$$M_{i,\boldsymbol{\alpha}} = \arg\min_{f \in \mathcal{F}} \mathbb{E}_{X,Y \sim P_i}[\mathcal{L}(Y, f(\boldsymbol{\alpha} \odot X))]$$

be the risk minimizer for population $i$ over some model class $\mathcal{F}$ when the covariates are masked by $\boldsymbol{\alpha}$. Accordingly, we can formalize our problem as

$$\min_{\boldsymbol{\alpha}} \max_{P_i \in \mathcal{P}} \mathbb{E}_{X,Y \sim P_i}[\mathcal{L}(Y, M_{i,\boldsymbol{\alpha}}(\boldsymbol{\alpha} \odot X))] \text{ s.t. } \|\boldsymbol{\alpha}\|_0 = k. \tag{1}$$

In order to solve this problem, we have access to samples $\{(X_i^j, Y_i^j)\}_{j=1}^{n_i}$ drawn iid for each population $P_i \in \mathcal{P}$.

## 3   Methods

There are two main challenges to solving this problem. First, the optimization over $\boldsymbol{\alpha}$ is discrete, since $\boldsymbol{\alpha}$ is binary – even for linear models and a single population, this is NP-hard [Natarajan, 1995]. We address this by introducing a continuous relaxation. Second, the population-level minimizer $M_{i,\boldsymbol{\alpha}}$ depends on $\boldsymbol{\alpha}$ in a nontrivial manner, as the solution to a risk minimization problem on the covariates included by $\boldsymbol{\alpha}$. In practice, we will only have finite data available to solve an empirical version of the risk minimization problem. Moreover, the model training process may induce complex

dependencies between the decision variable $\boldsymbol{\alpha}$ and the objective function, which is unlikely to have a closed form for general model families $\mathcal{F}$. To circumvent these issues, our method targets the loss of the *Bayes-optimal* predictor for each distribution and show that this surprisingly allows us to arrive at a significantly more computationally tractable optimization problem.

## 3.1 Continuous relaxation

While $\ell_0$-constrained problems are typically hard, we draw inspiration from the Lasso and relax to an $\ell_1$ constraint that restores convexity of the feasible set while still encouraging sparsity. In a continuous relaxation, $\boldsymbol{\alpha} \in [0,1]^m$ now gives the *degree* to which a variable is included instead of a binary decision. However, naively scaling inputs as $\boldsymbol{\alpha} \odot X$ allows flexible predictors to trivially undo the scaling. For example, if the model involves a linear transformation, it could internally learn a coefficient $w_i/\alpha_i$ for the input $\alpha_i X_i$. A deterministic scaling does not actually remove any information about the covariate unless $\alpha_i = 0$ exactly.

To avoid this issue, we introduce an alternative continuous relaxation that incorporates a *stochastic* component controlled by $\boldsymbol{\alpha}$; effectively, $\alpha_i$ will control the amount of noise added to the observation of $X_i$. Formally, let $\boldsymbol{\alpha} = (\alpha_1, \ldots, \alpha_m) \in \mathbb{R}^m_{\geq 0}$ be a vector of parameters controlling the degradation level for each covariate. We define the observed (noised) variable $S(\boldsymbol{\alpha})$ as a random variable whose $i$-th component $S_i(\boldsymbol{\alpha})$ distributed as

$$S_i(\boldsymbol{\alpha})|X \sim \mathcal{N}(X_i, \boldsymbol{\alpha}_i),$$

with the random variables independent across $i$ conditionally on $X$. Equivalently, $S(\boldsymbol{\alpha}) = X + \epsilon(\boldsymbol{\alpha})$, where $\epsilon(\boldsymbol{\alpha}) \sim \mathcal{N}(0, \Sigma_{\boldsymbol{\alpha}})$ with $\Sigma_{\boldsymbol{\alpha}} = \mathrm{diag}(\boldsymbol{\alpha}_1, \ldots, \boldsymbol{\alpha}_m)$. Here, $\boldsymbol{\alpha}_i = 0$ implies $S_i = X_i$ (no degradation), while $\boldsymbol{\alpha}_i \to \infty$ implies $S_i$ contains no information about $X_i$. Our goal is to find an $\boldsymbol{\alpha}$ vector that has small values for relevant features and large values for irrelevant ones, while maintaining predictive performance under distribution shifts. Formally, our objective becomes

$$\min_{\boldsymbol{\alpha}} \max_{P_i \in \mathcal{P}} \mathbb{E}_{S(\boldsymbol{\alpha}), Y \sim P_i}[\mathcal{L}(Y, \tilde{M}_{i,\boldsymbol{\alpha}}(S(\boldsymbol{\alpha})))] + \lambda \operatorname{Reg}(\boldsymbol{\alpha}) \tag{2}$$

$$\tilde{M}_{i,\boldsymbol{\alpha}} = \arg \min_{f \in \mathcal{F}} \mathbb{E}_{S(\boldsymbol{\alpha}), Y \sim P_i}[\mathcal{L}(Y, f(S(\boldsymbol{\alpha})))]. \tag{3}$$

so that we optimize the performance of the risk-minimizing models that observe the noisy version of the covariates. $\lambda$ is the regularization parameter which controls the sparsity of the solution, and can be varied to obtain a solution with the desired cardinality. We note that the regularization term here is in contrast to standard $\ell_1$ regularization, which typically promotes sparsity by shrinking irrelevant parameters toward zero, eg. we may choose $\operatorname{Reg}(\boldsymbol{\alpha}) = 1/\|\boldsymbol{\alpha}\|_1$.

## 3.2 Solving the relaxation

Directly solving the relaxation in Equation (3) is nontrivial. Each choice of $\boldsymbol{\alpha}$ leads to a different set of models $\tilde{M}_{i,\boldsymbol{\alpha}}$. Moreover, the structure of this mapping may be highly complex, mediated as it is by the inner minimization problem defining $\tilde{M}_{i,\boldsymbol{\alpha}}$. One strategy employed throughout the machine learning literature to solve problems with an inner optimization loop is to differentiate through the solution to the inner problem in order to optimize the outer objective via gradient descent [Amos and Kolter, 2017, Finn et al., 2017, Agrawal et al., 2019, Liu et al., 2018]. Intuitively, this corresponds to computing a gradient $\nabla_{\boldsymbol{\alpha}} \tilde{M}_{i,\boldsymbol{\alpha}}$ which captures how changes to $\boldsymbol{\alpha}$ in turn change the fitted model. Perhaps the closest analogy to our setting is model-agnostic meta-learning (MAML) [Finn et al., 2017] which solves meta-learning problems by differentiating through the training loops of models for individual tasks.

While it might be possible to adapt such a strategy to our setting, it would incur three key disadvantages.

1. **Computational Expense:** Retraining a potentially complex model $M$ for every adjustment to $\boldsymbol{\alpha}$ and for every considered $P_i$ during the optimization process is often prohibitively costly. Backpropagating through the model fitting process in every iteration is similarly costly.

2. **Optimization Instability:** Backpropagating gradients through the iterative training procedure of $M$ with respect to $\boldsymbol{\alpha}$ can be numerically unstable, suffer from vanishing/exploding gradients, or converge poorly, especially for deep or non-convex models.

3. **Specificity to a single model class:** Such a formulation would necessarily optimize for the performance of the particular class of models used to instantiate the inner loop. In order to maintain differentiability, this would likely have to be neural networks, even if other models (e.g., random forests) might be preferred for the tabular settings common in applications like medical risk prediction.

To circumvent these difficulties, we shift our focus from the performance of a specific, explicitly trained model $M$ to the performance of a Bayes-optimal optimal predictor, which represents the best possible performance achievable given the selected features. We believe this to be a good target for optimization for multiple reasons. Firstly, it represents a model-family agnostic target which will be more closely approached if a practitioner makes good choices for the modeling approach within any specific setting and distribution. Secondly, in many settings, after feature selection, we might be able to collect a substantial amount of new data corresponding to the chosen covariates. E.g., consider a health system that decides on new survey items to gather based on a pilot study, and can then observe new data from the routine deployment of the selected questions. In such a setting, the Bayes-optimal loss may be a better proxy for the long-term performance of the system. Third, starting from the perspective of the Bayes-optimal predictor will allow us to further simplify the objective and arrive at a significantly more computationally tractable approach. In the following sections, we formalize this approach and show how it leads to a simple closed-form objective.

**Population-level objective**  We now express the population-level formulation of this problem using the Bayes-optimal predictor for $Y$ given $S(\boldsymbol{\alpha})$, denoted by $f^*(S(\boldsymbol{\alpha})) = \mathbb{E}[Y|S(\boldsymbol{\alpha})]$. With respect to the MSE, the expected loss of this optimal predictor is the conditional variance of $Y$ given $S(\boldsymbol{\alpha})$ (since the bias term is 0):

$$\mathbb{E}_{(S(\boldsymbol{\alpha}),Y)\sim P_i}[(Y - \mathbb{E}[Y|S(\boldsymbol{\alpha})])^2] = \mathbb{E}_{S(\boldsymbol{\alpha})\sim P_i}[\mathbb{V}[Y|S(\boldsymbol{\alpha})]]$$

Applying the law of total variance and dropping terms constant in $\boldsymbol{\alpha}$ leads to an equivalent optimization problem that depends only on the conditional variance of $\mathbb{E}[Y \mid X]$ given $S(\boldsymbol{\alpha})$. This can be formalized as follows.

**Theorem 1** (Population-Level Objective). *Under the noise-based relaxation $S(\boldsymbol{\alpha}) = X + \epsilon(\boldsymbol{\alpha})$, $\epsilon(\boldsymbol{\alpha}) \sim \mathcal{N}(0, \mathrm{diag}(\boldsymbol{\alpha}))$, the distributionally robust feature selection problem is equivalent to*

$$\min_{\boldsymbol{\alpha}} \max_{P_i \in \mathcal{P}} -\mathbb{E}_{S(\boldsymbol{\alpha})\sim P_i}\big[\mathbb{E}_{X\sim P_i}[\mu_i(X) \mid S(\boldsymbol{\alpha})]^2\big] + \lambda \operatorname{Reg}(\boldsymbol{\alpha}),$$

*where $\mu_i(X) = \mathbb{E}_{P_i}[Y \mid X]$.*

The proof of Theorem 1 is provided in Appendix A.1.

### 3.3 Empirical estimation and kernel form of objective

So far, we have dealt only with population-level quantities. Next, we must develop a strategy to estimate these using the sampled data that we observe. Given samples $\{(X_i^j, Y_i^j)\}$ from each population $P_i$, let $\hat{\mu}_i(X)$ be an estimator of $\mu_i(X) = \mathbb{E}[Y|X]$ trained on these samples. The empirical form of the objective from Theorem 1 is

$$\min_{\boldsymbol{\alpha}} \max_{P_i \in \mathcal{P}} -\widehat{\mathbb{E}}_{S(\boldsymbol{\alpha})\sim P_i}\big[\widehat{\mathbb{E}}[\hat{\mu}_i(X) \mid S(\boldsymbol{\alpha})]^2\big] + \lambda \operatorname{Reg}(\boldsymbol{\alpha}), \tag{4}$$

where the expectation $\widehat{\mathbb{E}}[\hat{\mu}_i(X) \mid S(\boldsymbol{\alpha})]^2\big]$ is taken with respect to the empirical distribution $\hat{P}_i(X) = \frac{1}{n_i}\sum_{j=1}^{n_i} \delta_{X_i^j}(X)$. First, we propose to estimate $\mu_i(X)$ simply by fitting a machine learning model to the samples $\{(X_i^j, Y_i^j)\}$ from population $i$. This can be done just once, using an arbitrary model well-suited to the application domain. There will be no need to differentiate through the training process, or refit the model during training. To make the empirical objective in Equation (4) computationally tractable, we express the conditional expectation $\widehat{\mathbb{E}}[\hat{\mu}_i(X) \mid S(\boldsymbol{\alpha})]$ in closed form using Bayes' theorem under the Gaussian noise model. This yields a kernel-weighted representation of $\mu_i(X)$, where each observed sample contributes according to its likelihood under the noisy observation $S(\boldsymbol{\alpha})$.

**Theorem 2** (Kernel Form Equivalence). *The empirical expectation*

$$\widehat{\mathbb{E}}_{S(\boldsymbol{\alpha})} \left[ \widehat{\mathbb{E}}[\hat{\mu}_i(X) \mid S(\boldsymbol{\alpha})]^2] \right]$$

*is equivalent to*

$$\widehat{\mathbb{E}}_{S(\boldsymbol{\alpha})}\left[\left(\sum_{j=1}^{n_i} w_i^j(S, \boldsymbol{\alpha})\hat{\mu}_i(X_i^j)\right)^2\right]$$

*where the weights are*

$$w_i^j(S, \alpha) = \frac{\exp\left(-\frac{1}{2}\left(X_i^j - S(\boldsymbol{\alpha})\right)^T \text{diag}(\alpha)^{-1}\left(X_i^j - S(\boldsymbol{\alpha})\right)\right)}{\sum_{k=1}^{n_i}\exp\left(-\frac{1}{2}\left(X_i^k - S(\boldsymbol{\alpha})\right)^T \text{diag}(\alpha)^{-1}\left(X_i^k - S(\boldsymbol{\alpha})\right)\right)}$$

The proof for Theorem 2 is provided in Appendix A.2. This equivalence shows that the conditional expectation can be estimated via Gaussian kernel smoothing, where the bandwidth of the kernel is determined by $\boldsymbol{\alpha}$. In summary, this kernel-based formulation provides an explicit, differentiable link between the feature-noise parameters $\boldsymbol{\alpha}$ and the resulting predictive uncertainty. It allows us to efficiently approximate the objective using empirical samples, enabling the end-to-end optimization procedure described in the following section.

### 3.4 Algorithmic approach

The final objective function takes an expectation of this over the random draw of $S$ itself. Accordingly, the final statement of the optimization problem we wish to solve is

$$\min_{\boldsymbol{\alpha}} \max_{P_i \in \mathcal{P}} -\mathbb{E}_{S(\boldsymbol{\alpha})}\left[\left(\sum_{j=1}^{n_i} w_i^j(S, \boldsymbol{\alpha})\mu_i(X_i^j)\right)^2\right].$$

We implement gradient descent algorithm for this problem. We draw samples of $S$ to approximate the outer objective function. Specifically, for $b$ Monte Carlo samples $S^1...S^b$, we obtain the objective

$$-\frac{1}{b}\sum_{\ell=1}^{b}\left(\sum_{j=1}^{n_i} w_i^j(S^\ell, \boldsymbol{\alpha})\mu_i(X_i^j)\right)^2.$$

We take a gradient descent step with respect to either the maximum loss over all the populations, or the softmax over the population losses, controlled by the temperature parameter $\beta$. Since the distribution of $S(\boldsymbol{\alpha})$ itself depends on $\boldsymbol{\alpha}$, we construct each sample via the reparameterization trick: $S = X + \sqrt{\boldsymbol{\alpha}} \odot \epsilon$, where $\epsilon \sim \mathcal{N}(0, I)$. This ensures that gradients can flow not only through the kernel weights $w_i^j(S, \boldsymbol{\alpha})$, but also through the sampled values $S$ themselves, enabling end-to-end differentiation of the entire objective. This can now be implemented entirely with standard autodifferentiation software since $w$ is a closed-form, differentiable function of $\boldsymbol{\alpha}$. All that is required is to fit a model $\mu_i(X)$ once at the start of the process for each population; it can then be reused at each iteration unchanged and we never need to differentiate through the model training process. In practice, we further improve computational efficiency by taking the inner sum only over the $k$-nearest neighbors of $S^{(\ell)}$ in the set $\{X_i^j\}_{j=1}^{n_i}$ since the conditional distribution of $X$ given $S$ typically places a negligible mass outside this set.

The algorithm concludes by selecting the $k$ features with the smallest optimized noise parameters $\alpha^*$, as these represent the most informative variables for maintaining prediction performance across all populations. We summarize our complete proposed method in Algorithm 1 in Appendix B, and provide its computational complexity.

## 4 Experiments

We conduct experiments on both synthetic and real-world datasets. We evaluate our proposed method against baseline models trained on a pooled dataset comprising all populations. Specifically, we use Lasso regression (linear regression for real targets, and logistic regression for categorical) and XGBoost [Chen and Guestrin, 2016] regression/classification as baselines. For Lasso, feature

selection is based on the largest coefficients under the regularization path, while for XGBoost, we rely on internal feature importance scores. We also implement distributionally robust (DRO) variants of both models. Further, we include an embedded baseline (Embedded MLP), which uses a multi-layer-perceptron (MLP) with a learnable feature mask trained via DRO. A comprehensive description of all baseline methods can be found in Appendix C. For the downstream prediction task, we train a model on the selected features. The downstream model training is carried out independently for each feature selection method. We implemented our method using the `PyTorch` [Paszke et al., 2019] library, while for the downstream models, we use the `scikit-learn` [Pedregosa et al., 2011] library. All baselines plus our method share the pipeline for downstream models, isolating the impact of the feature selections they output as opposed to predictive performance of models that they use en route.

**Data processing** We split each dataset into three parts – a *feature-selection-dataset*, *downstream-model-training-dataset* and a *downstream-model-test-dataset*. We first do a $60 : 40$ split of each population to obtain the *feature-selection-dataset* and the downstream model training and evaluation datasets. The latter is split $80 : 20$ for downstream-model training and evaluation respectively.

**Evaluation Metrics** In keeping with our motivation to improve downstream prediction accuracy, we first perform feature selection using our method and the baseline methods. We then train the downstream model independently on the *downstream-model-training-dataset*, using the features selected by each method. We then evaluate performance of the fitted downstream models (random forest and MLP) on the *downstream-model-test-dataset* using the mean-squared-error (MSE) as the primary metric for regression datasets, and Log Loss on the classification dataset. For the real datasets, we also include $R^2$-score for regression and prediction accuracy for classification tasks. The tabulated metrics are provided in Appendix D.

## 4.1 Synthetic Data Experiments

To systematically evaluate the behavior of our method under controlled settings, we conduct a series of synthetic experiments that vary in functional form, dimensionality, and population heterogeneity. The first experiment considers a purely linear data-generating process with population-specific coefficients to study the effect of structured covariate shifts. A higher-dimensional variant of this experiment is provided in Appendix E.1. The second experiment introduces nonlinear relationships and heterogeneous noise distributions across populations, allowing us to assess robustness to more complex, mixed effects. Finally, an additional experiment using a sparse linear prediction model for targets is included in Appendix E.2. For the downstream prediction models, we use a random forest and an MLP– both models use the standard scikit-learn implementations for fitting to the output. The random forest is an ensemble of 100 decision trees, while the MLP has a single hidden layer with 100 neurons, trained for a maximum of 1000 epochs with early stopping based on validation loss convergence (handled internally by `scikit-learn`).

**Synthetic dataset 1: Linear model** This synthetic dataset comprises three populations $A, B, C$, with 15 features each. The populations have the following proportions in the dataset, and (linear) outcome functions:

$$
\begin{aligned}
&\text{A (40\%)} && Y = 8X_0 + 6X_1 - 4X_2 + 3X_3 + 2X_4 + \epsilon \\
&\text{B (35\%)} && Y = -8X_0 - 6X_1 + 4X_2 - 3X_3 - 2X_4 + 8X_5 + 6X_6 + \epsilon \\
&\text{C (25\%)} && Y = 10X_7 + 8X_8 + 6X_9 - 5X_{10} + \epsilon \\
&\text{Noise} && \epsilon \sim \mathcal{N}(0, 0.1^2)
\end{aligned}
$$

We set the budget for feature selection to $5$ and, for comparison, also include the results for an increased budget of $10$. Figure 1 shows the performance of the downstream prediction models on the task of predicting $Y$ for each population, using the features selected using our method and the baselines. The complete metrics table can be found in Appendix D, Table 1. More details on the implementation and hyperparameters used are provided in Appendix D.1. We also include an additional experiment with this data set in which the number of features increases to $50$, the results of which can be found in Appendix E.1.

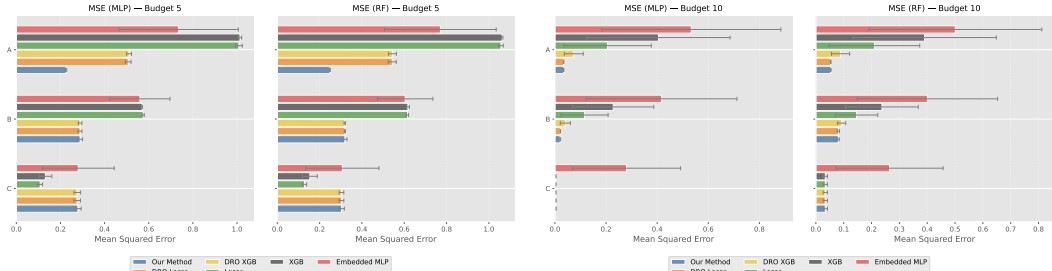

(a) **Budget = 5**: Our method consistently achieves low error across all populations, performing on par with DRO-XGBoost and DRO-Lasso, even outperforming both methods in population $A$.

(b) **Budget = 10**: Overall performance improves predictably upon increasing the feature budget. Our method maintains lower MSE compared to other methods, including DRO-Lasso.

Figure 1: **Performance comparison across populations on synthetic dataset 1 using different feature selection methods**. In each subplot, left: Mean Squared Error of downstream MLP model; right: Mean Squared Error of downstream random forest model. The relative performance of feature selection methods is consistent across the choice of downstream prediction model.

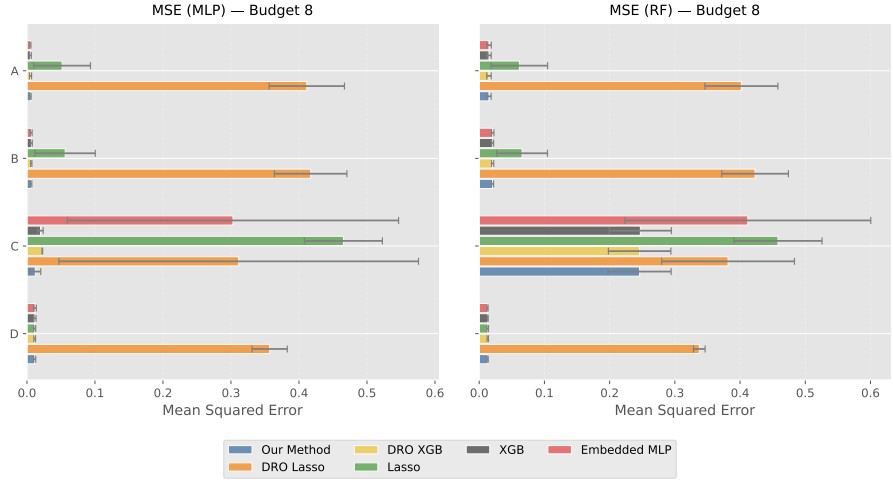

Figure 2: **Performance comparison across populations on synthetic dataset 2**. Left: Mean Squared Error of downstream MLP model; right: Mean Squared Error of downstream random forest (RF) model. Our method consistently achieves low error across populations and budgets, outperforming or matching DRO-based baselines.

**Synthetic experiment 2: Linear model** Here we have four populations, $A, B, C, D$, with 50 features each. The populations have the following proportions in the dataset, and outcome functions:

$$A\ (40\%) \quad Y = 4X_0 + 3X_1 + X_2^2 + \epsilon_A$$
$$B\ (35\%) \quad Y = 4X_0 + 3X_1 + X_2^2 + \epsilon_B$$
$$C\ (25\%) \quad Y = 2X_0 + 3X_5X_6 + 4\sin(2X_7) + \epsilon_C$$
$$D\ (15\%) \quad Y = 3X_0 + 2X_1 + \epsilon_D$$

with heterogeneous noise across populations: Population $A$ experiences reduced noise with $\epsilon_A \sim \mathcal{N}(0, 0.05^2)$, while Population $B$ exhibits heteroscedasticity where $\epsilon_B = \sigma(X_3, X_4) \cdot \eta \cdot 0.1$, with $\sigma = \exp(0.5X_3 + 0.3X_4)$ and $\eta \sim \mathcal{N}(0, 1)$. Population $C$ follows a standard noise model $\epsilon_C \sim \mathcal{N}(0, 0.1^2)$, and Population $D$ is characterized by heavy-tailed noise drawn from a scaled $t$-distribution: $\epsilon_D \sim t_3 \cdot 0.2$.

We set the budget for feature selection to 8. Figure 2 shows the performance of the downstream prediction models on the task of predicting $Y$ for each population, using the features selected using

our method and the baselines. The complete metrics table can be found in Appendix D, Table 2. More details on the implementation and hyperparameters used are provided in Appendix D.2.

**Results**   In *synthetic experiment 1*, although the generative process is linear and variables $X_0$ to $X_4$ have strong effects in both populations $A$ and $B$, the signs of their coefficients are reversed between populations. This reduces the effectiveness of LASSO, which tends to select features based on average effects across all data. As a result, vanilla LASSO achieves its best performance on population $C$, as features relevant for $C$ have no conflicting coefficients, but performs poorly on $A$ and $B$, in spite of the linear setting. Vanilla XGBoost shows a similar trend. Our method outperforms most baselines, and has a balanced performance across populations, not performing inordinately well in a certain population at the cost of others, unlike vanilla Lasso and XGBoost. For budget=10, our method is comparable with the best performing baselines, namely DRO XGB and DRO Lasso. The Embedded MLP baseline performs poorly even for the higher budget.

In *synthetic experiment 2*, the nonlinear data setting puts both vanilla Lasso and its DRO variant at a disadvantage, as seen in Figure 2, with DRO Lasso having the worst performance. The Embedded MLP baseline also performs poorly, being imbalanced in favor of other populations at the cost of population $C$, similar to Lasso. Our method is again comparable with the best performing baselines of XGB (DRO and vanilla), for both budgets.

It should also be noted that our method has consistently low variance across both the synthetic experiments, the exact values of which may be found in Tables 1 and 2. The relative performance of feature selection methods is consistent across the choice of downstream prediction model.

## 4.2   Real-datasets

**UCI Adult Income Dataset [Becker and Kohavi, 1996]**   We use the UCI Adult Income dataset to predict income across different demographic groups, where each age group represents a distinct population. The dataset contains census income data with 14 features including age, education, occupation, and demographic information. The target variable is binary, indicating whether an individual's income exceeds \$50K per year. After combining the original train/test splits ($\approx 48,000$ samples), we subsample $10\%$ of the data for faster computation. Populations are defined by sex (Male/Female). We drop any features used to define the population group. After one-hot-encoding, we have $44$ features, of which we select $5$ (i.e. we set our budget to $5$). Since the task is classification, a Random Forest Classifier is trained on selected features. Training is performed on each population separately, using each set of selected features. For further details about hyperparameter choices, please see Appendix D.4.

**American Community Survey (ACS) Dataset [U.S. Census Bureau, 2018]**   We use person-level ACS Public Use Microdata Sample (PUMS) data for the year 2018 to predict household income across state populations. We focus on three populations, namely California (CA), FLorida (FL) and New York (NY). We subsample $5\%$ of each state's population for faster computation. To further ease computation load, we also subsample features by fitting a random forest regressor to the entire pooled dataset of all the populations, and picking the top 18 most important features. We then perform feature selection using our method, and the baselines. For the downstream evaluation, since the target is real valued (similar to the synthetic datasets), we fit a Random Forest Regressor using the selected features. For further details about hyperparameter choices, please see Appendix D.3.

**Data pre-processing**   Categorical features are one-hot encoded to create numerical representations. Missing values are imputed using median imputation. Any instances with missing target values are dropped. The numerical features are standardized to ensure consistent scale across features. From the ACS data, we drop all non-numeric or weight/geoid features, and retain only the numeric features.

## 4.3   Results

Our method consistently outperforms all baseline approaches across both the ACS and UCI datasets. On the ACS regression task, we observe (Figure 3a) an order-of-magnitude reduction in MSE and substantial gains in $R^2$ across all populations. This indicates that our feature selection procedure identifies highly informative variables. Notably, even strong baselines such as DRO-XGBoost suffer from significantly higher error and variance, suggesting sensitivity to spurious or less transferable

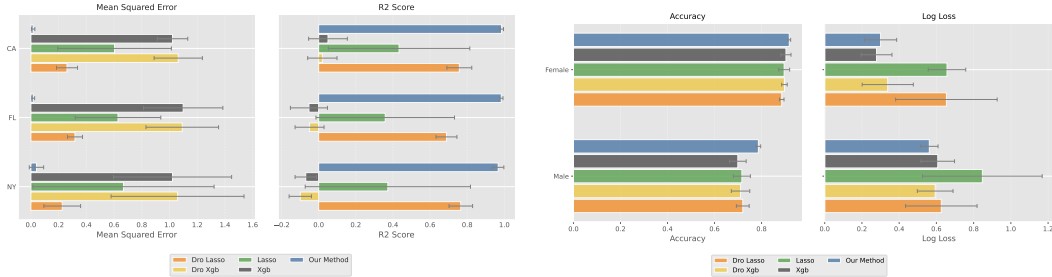

(a) **ACS dataset results**: Comparison of mean squared error (MSE) and $R^2$ score (with standard deviations) for each method across California (CA), Florida (FL), and New York (NY). Our method achieves an order-of-magnitude lower MSE and substantially higher $R^2$ than all baselines, with consistently low run-to-run variance.

(b) **UCI dataset results**: Downstream classification performance (mean accuracy and log loss with standard deviations) for each method on Female and Male populations. Our method achieves the highest accuracy in both groups while maintaining a comparative log loss and generally lower run-to-run variance.

Figure 3: **Performance comparison across populations on real datasets using different feature selection methods**. In each subplot, left: Mean Squared Error; right: $R^2$ Score.

features across groups. We also note that on the ACS dataset, the Lasso variants outperform XGBoost. For exact metric values, please see Table 3 in Appendix D.3.

On the UCI classification task, our method generally outperforms the baselines across both populations, with the exception of the Log Loss over the Female population, where XGBoost outperforms it (Figure 3b). However, our method has a more balanced performance across the two population. Furthermore, the consistently low standard deviation across all reported metrics demonstrates the stability of our method across random initializations and training splits. For exact metric values, please see Table 4 in Appendix D.4.

## 5 Discussion and future work

We have presented a novel noise-based continuous relaxation framework for distributionally robust feature selection in a group-DRO setting. In contrast to existing methods that optimize for average-case performance or train a single robust model, our approach directly targets the selection of features that enable high-quality models across multiple subpopulations. By injecting feature-wise noise and optimizing the Bayes-optimal predictor's variance, we derive a model-agnostic and computationally tractable objective that avoids differentiating through model training. This formulation allows us to identify features with stable predictive utility under distribution shifts.

Empirical results highlight the limitations of standard selection techniques—such as Lasso—in capturing non-linear or population-specific signal. Our method achieves substantially improved performance, including an order-of-magnitude reduction in MSE on the ACS dataset, underscoring its practical value in real-world settings.

In future work, we aim to apply our method to a broader range of real-world datasets to further assess its generalizability and practical utility. Replacing our current plug-in estimators with influence function-based approaches could reduce estimation bias, especially when dealing with limited samples from minority populations. This would improve the robustness of our feature selection when population sizes are imbalanced.

Additionally, extending our framework beyond MSE loss presents an interesting theoretical challenge. Our derivation leverages the bias–variance decomposition specific to mean squared error, though generalized variance decompositions have been proposed for broader classes of loss functions, including proper scoring rules such as cross-entropy. Incorporating these generalized forms could allow our method to directly optimize alternative objectives, though this would require nontrivial extensions of the current theoretical framework. Empirically, our results with cross-entropy loss in the UCI experiment suggest that optimizing MSE within our framework transfers well to other proper scoring losses, indicating that our bias–variance-based reasoning may extend beyond MSE in practice. Developing a theoretical foundation for such extensions is an important direction for future work.

## Acknowledgments

Research reported in this publication was supported by the National Institute of Mental Health of the National Institutes of Health under award number R01MH139097, and the AI Research Institutes Program funded by the National Science Foundation under AI Institute for Societal Decision Making (AI-SDM), Award No. 2229881.

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

# A Theoretical Results: Derivations and Intuition

In this appendix, we provide detailed derivations and additional intuition for the theoretical results presented in the main text. We begin by expanding the population-level formulation of the objective, clarifying its interpretation in terms of conditional variances. We then derive the equivalent kernel-based form used in the algorithmic implementation.

## A.1 Population-level objective

In this section, we derive the population-level objective in Theorem 1. Recall that the Bayes-optimal predictor for $Y$ given $S(\boldsymbol{\alpha})$, is given by $f^*(S(\boldsymbol{\alpha})) = \mathbb{E}[Y|S(\boldsymbol{\alpha})]$. With respect to the MSE, the expected loss of this optimal predictor is the conditional variance of $Y$ given $S(\boldsymbol{\alpha})$ (since bias is 0):

$$\mathbb{E}_{(S(\boldsymbol{\alpha}),Y)\sim P_i}[(Y - \mathbb{E}[Y|S(\boldsymbol{\alpha})])^2] = \mathbb{E}_{S(\boldsymbol{\alpha})\sim P_i}[\mathbb{V}[Y|S(\boldsymbol{\alpha})]]$$

We can expand this using the law of total variance to rewrite

$$\mathbb{V}[Y|S(\boldsymbol{\alpha})] = \mathbb{E}_X[\mathbb{V}[Y|S(\boldsymbol{\alpha}), X]|S(\boldsymbol{\alpha})] + \mathbb{V}[\mathbb{E}[Y|S(\boldsymbol{\alpha}), X]|S(\boldsymbol{\alpha})]. \tag{5}$$

Given the generative process $S(\boldsymbol{\alpha}) = X + \boldsymbol{\alpha}\sqrt{\epsilon}$, where the noise $\epsilon$ is independently sampled, we have $Y \perp\!\!\!\perp S|X$ and thus can drop $S(\boldsymbol{\alpha})$ from the inner conditioning terms. We obtain:

$$\mathbb{V}[Y|S(\boldsymbol{\alpha})] = \mathbb{E}_X[\mathbb{V}[Y|X]|S(\boldsymbol{\alpha})] + \mathbb{V}[\mathbb{E}[Y|X]|S(\boldsymbol{\alpha})]. \tag{6}$$

This expression has the appealing property that $\boldsymbol{\alpha}$ only enters through the outer conditioning: it changes the distribution over which we take the expectation/variance, but not the variable we are taking the expectation/variance of. Thus we may estimate $\mathbb{V}[Y|X]$ and $\mathbb{E}[Y|X]$ just once per test distribution $P_i$, instead of having to fit a new model for every value $\boldsymbol{\alpha}$. However, we can pursue further simplifications to arrive at an even more streamlined objective with closed-form dependence on $\boldsymbol{\alpha}$.

First, recall that our objective is to minimize the expected variance $\mathbb{E}_{S(\boldsymbol{\alpha})}[\mathbb{V}[Y|S(\boldsymbol{\alpha})]]$. When wrapped in this outer expectation, the first term in Equation 6 above becomes constant with respect to $\boldsymbol{\alpha}$: $\mathbb{E}_{S(\boldsymbol{\alpha})}[\mathbb{E}_X[\mathbb{V}[Y|X]|S(\boldsymbol{\alpha})]] = \mathbb{E}_X[\mathbb{V}[Y|X]]$. Accordingly, we can drop it from the optimization objective. We are left with the problem

$$\min_{\boldsymbol{\alpha}} \max_{P_i \in \mathcal{P}} \mathbb{E}_{S(\boldsymbol{\alpha})\sim P_i}[\mathbb{V}[\mathbb{E}[Y|X]|S(\boldsymbol{\alpha})]] \tag{7}$$

Intuitively the inner variance term $\mathbb{V}[\mathbb{E}[Y|X]|S(\boldsymbol{\alpha})]$ represents the variance of the true conditional expectation $\mathbb{E}[Y|X]$ conditioned on us observing $S(\boldsymbol{\alpha})$. Specifically, given $S(\boldsymbol{\alpha}) = s$, there is a distribution of possible true covariate $(X)$ values that could have generated the corresponding $S(\boldsymbol{\alpha})$ through the noising process. Each of these potential $X$ values has an associated true conditional mean $\mathbb{E}[Y|X]$. The term $\mathbb{V}[\mathbb{E}[Y|X]|S(\boldsymbol{\alpha}) = s]$ measures the variability or *spread* of these true $\mathbb{E}[Y|X]$ values, given the observed $s$. A high value of this conditional variance indicates that a single noisy observation is consistent with a wide range of possible values for $X$, and correspondingly, $\mathbb{E}[Y|X]$ values. In this scenario, the noise introduced by $\boldsymbol{\alpha}$ has significantly obscured the relationship between the observed $S(\boldsymbol{\alpha})$ and the underlying true predictive signal $\mathbb{E}[Y|X]$. Conversely, a low value for this variance implies that $S(\boldsymbol{\alpha})$ is highly informative about $\mathbb{E}[Y|X]$. Our objective, therefore, seeks to choose noise levels $\boldsymbol{\alpha}$ such that, even under the worst-case test distribution, the average uncertainty (variance) about the true predictive function $\mathbb{E}[Y|X]$ given the noised observation $S(\boldsymbol{\alpha})$ is minimized. In order to further simplify the objective we expand the variance term above. Let $\mu_i(X) = \mathbb{E}_{P_i}[Y|X]$ denote the conditional mean function on the $i$th population. We have

$$\mathbb{E}_{S(\boldsymbol{\alpha})\sim P_i}[\mathbb{V}[\mu_i(X)|S(\boldsymbol{\alpha})]] = \mathbb{E}_{S(\boldsymbol{\alpha})\sim P_i}[E_X[\mu_i(X)^2|S(\boldsymbol{\alpha})]] - \mathbb{E}_{S(\boldsymbol{\alpha})\sim P_i}[\mathbb{E}[\mu_i(X)|S(\boldsymbol{\alpha})]^2]$$

where the first term again collapses to $\mathbb{E}[\mu_i(X)^2]$, which is constant with respect to $\boldsymbol{\alpha}$ and can be dropped from the optimization. We are left with the second term, $-\mathbb{E}_{S(\boldsymbol{\alpha})\sim P_i}[\mathbb{E}[\mu_i(X)|S(\boldsymbol{\alpha})]^2]$. This term requires us to average $\mu(X_i)$ over the conditional distribution of $X$ given $S$.

## A.2 Kernel Form Equivalence Proof

In this section, we derive the kernel-form of the objective in Theorem 2. We need to estimate the conditional expectation over $\mu_i(X)$. By applying Bayes theorem, we can arrive at an estimate for

this expression with a closed form in terms of $\boldsymbol{\alpha}$. Specifically, we can rewrite

$$\mathbb{E}[\mu_i(X)|S] = \int \mu_i(X)\, dP_i(X|S(\boldsymbol{\alpha}))$$

where $\mu$ is averaged over the distribution of $X$ implied by the noise model, $P_i(X|S(\boldsymbol{\alpha}))$. Via Bayes theorem, we can rewrite

$$P_i(X|S(\boldsymbol{\alpha})) = \frac{P_i(X)P_i(S(\boldsymbol{\alpha})|X)}{P_i(S(\boldsymbol{\alpha}))}.$$

We propose to estimate this quantity, and hence the objective function, by setting the "prior" $P_i(X)$ in this expression to be the uniform distribution over the observed samples of $X$ from population $P_i$, $X_i^1...X_i^{n_i}$. The likelihood $P_i(S(\boldsymbol{\alpha})|X)$ under our Gaussian noise model for $S$ is given simply by

$$P_i(S(\boldsymbol{\alpha})|X) \propto \exp\left\{ -\frac{1}{2}(X - S)^T \operatorname{diag}(\boldsymbol{\alpha})^{-1}(X - S)\right\}$$

and we can then calculate the denominator as $\frac{1}{n_i}\sum_{j=1}^{n_i} P_i(S(\boldsymbol{\alpha})|X_i^j)$. Putting this all together, let

$$w_i^j(S, \boldsymbol{\alpha}) = \frac{P_i(S(\boldsymbol{\alpha})|X_i^j)}{\sum_{j=1}^{n_i} P_i(S(\boldsymbol{\alpha})|X_i^j)}$$

denote the estimate of the probability of observed data point $X_i^j$. We arrive at the estimator

$$\widehat{\mathbb{E}}[\mu_i(X)|S]^2 = \left( \sum_{j=1}^{n_i} w_i^j(S, \boldsymbol{\alpha})\mu_i(X_i^j)\right)^2$$

which can be interpreted as kernel smoothing of $\mu_i(X)$ under the Gaussian kernel implied by the noise model, measuring the amount of information lost due to the smoothing.

## B  DRO Feature Selection Algorithm and complexity

In this section, we summarize the full procedure for optimizing the noise-based relaxation and obtaining the final set of robust features described in section 3. We present the algorithmic implementation of our method, followed by an analysis of its computational complexity. The proposed procedure, is outlined in algorithm 1.

**Computational complexity per iteration**  Our method requires $O(P \cdot b \cdot n \cdot K \cdot d)$ operations per iteration, where $P$ is the number of populations, $b$ is the number of Monte Carlo samples, $n = \max_p n_p$ is the maximum population size, $K$ is the number of nearest neighbors used for kernel weight computation, and feature dimensionality $d$.

## C  Baseline Methods

**Data Pooling & Standardization:** Data points $(X_i^{(p)}, Y_i^{(p)})$ from all populations $p = 1, \ldots, P$ are pooled into a single dataset $\{(X_j, Y_j)\}_{j=1}^N$. Both features $X$ and outcomes $Y$ are then standardized to have zero mean and unit variance.

We summarize the baseline methods below:

1. **Vanilla Lasso regression**
2. **Vanilla XGBoost regression**
3. **DRO Lasso regression**
4. **DRO Lasso regression**
5. **Embedded MLP**

---

**Algorithm 1** Distributionally Robust Feature Selection

---

**Input:** Dataset $\{(X_j^{(p)}, Y_j^{(p)})\}_{j=1}^{n_p}$ for populations $p = 1, \ldots, P$; budget $k$; regularization parameter $\lambda$; learning rate $\eta$; Monte Carlo samples $b$; number of nearest neighbors $K$

**Output:** Feature noise parameters $\boldsymbol{\alpha}^* \in \mathbb{R}_{\geq 0}^m$

 1: **Precompute:** For each population $p$:
 2:      Fit model $\hat{\mu}_p(X) \approx \mathbb{E}[Y|X]$ using $\{(X_j^{(p)}, Y_j^{(p)})\}_{j=1}^{n_p}$
 3:      Build k-NN index for $\{X_j^{(p)}\}_{j=1}^{n_p}$                    $\triangleright$ Optional: for efficiency
 4: **Initialize:** $\boldsymbol{\alpha}^{(0)} \leftarrow \mathbf{1} + \epsilon$, where $\epsilon \sim \mathcal{N}(0, \sigma^2 I)$
 5: **for** epoch $= 1$ **to** $T_{\max}$ **do**
 6:      $\mathcal{L} \leftarrow 0$
 7:      **for** $p = 1$ **to** $P$ **do**
 8:          $\mathcal{L}_p \leftarrow 0$
 9:          **for** $\ell = 1$ **to** $b$ **do**
10:              Sample noise $\xi \sim \mathcal{N}(0, I)$
11:              Generate noisy observation $S^{(\ell)} \leftarrow X + \sqrt{\boldsymbol{\alpha}^{(\text{epoch}-1)}} \odot \xi$
12:              **if** using k-NN **then**
13:                  $\mathcal{N} \leftarrow$ k-nearest neighbors of $S^{(\ell)}$ in $\{X_j^{(p)}\}_{j=1}^{n_p}$
14:              **else**
15:                  $\mathcal{N} \leftarrow \{1, \ldots, n_p\}$
16:              **for** $j = 1$ **to** $n_p$ **do**
17:                  Compute weights:
18:
$$w_j^p(S^{(\ell)}, \boldsymbol{\alpha}) \leftarrow \frac{\exp\left(-\frac{1}{2}(X_j^{(p)} - S^{(\ell)})^T \text{diag}(\boldsymbol{\alpha})^{-1}(X_j^{(p)} - S^{(\ell)})\right)}{\sum_{j' \in \mathcal{N}} \exp\left(-\frac{1}{2}(X_{j'}^{(p)} - S^{(\ell)})^T \text{diag}(\boldsymbol{\alpha})^{-1}(X_{j'}^{(p)} - S^{(\ell)})\right)}$$
19:              $\mathcal{L}_p \leftarrow \mathcal{L}_p - \frac{1}{b}\left(\sum_{j=1}^{n_p} w_j^p(S^{(\ell)}, \boldsymbol{\alpha})\hat{\mu}_p(X_j^{(p)})\right)^2$
20:      $\mathcal{L} \leftarrow \max_{p \in \{1, \ldots, P\}} \mathcal{L}_p + \lambda \cdot \text{Reg}(\boldsymbol{\alpha})$             $\triangleright$ or use softmax
21:      $\boldsymbol{\alpha}^{(\text{epoch})} \leftarrow \boldsymbol{\alpha}^{(\text{epoch}-1)} - \eta \nabla_{\boldsymbol{\alpha}} \mathcal{L}$
22:      Project $\boldsymbol{\alpha}^{(\text{epoch})}$ to $\mathbb{R}_{\geq 0}^m$                  $\triangleright$ Ensure non-negativity
23: **Feature Selection:** Select $k$ features with smallest $\boldsymbol{\alpha}^*$ values
24: **return** Selected feature indices $\mathcal{I} = \{\text{indices of } k \text{ smallest } \alpha_i^*\}$

---

**Vanilla Lasso**    This method applies Lasso regression to the combined dataset from all populations. The Lasso model is trained by solving:

$$\min_{\beta \in \mathbb{R}^d} \frac{1}{2N} \sum_{j=1}^{N} (Y_j - X_j^T \beta)^2 + \lambda_L ||\beta||_1$$

We set $\lambda_L$ value to 0.01. This selection is performed by retraining the model for each $\lambda_L$ and evaluating its performance on the full training set. Once the coefficients $\beta^*$ are determined, the $k$ features with the largest absolute coefficient values ($|\beta_i^*|$) are selected.

**Vanilla XGBoost**    This method utilizes feature importance scores from an XGBoost model trained on the pooled dataset. Hyperparameters such as tree depth, learning rate, etc., are set to default values provided by the XGBoost library[Chen and Guestrin, 2016]. Feature importance scores are extracted from the trained XGBoost model using the default `feature_importances_` attribute). The $k$ features with the highest importance scores are selected.

**Distributionally Robust Optimization (DRO) Lasso**    This method adapts Lasso to be robust by iteratively re-weighting populations to focus on worst-case performance. Features $X^{(p)}$ and outcomes $Y^{(p)}$ are standardized separately for each population $p$. We then perform **DRO Lasso**, summarized in Algorithm 2.

The $k$ features with the largest absolute coefficient values from the final Lasso model $\beta^{(T_{max})}$ are selected.

---
**Algorithm 2** DRO Lasso
---

1: **Initialize** uniform weights $w_p^{(0)} = \frac{1}{P}$ for each population $p = 1, \ldots, P$.
2: **Standardize** each population's data $(X^{(p)}, Y^{(p)})$ independently.
3: **Select** L1 regularization parameter $\lambda_L$
4: **for** $t = 1 \ldots T_{\max}$ **do**
5:     **Construct Pooled Dataset with Sample Weights:**
       Form $X_{\text{all}}, Y_{\text{all}}$ by concatenating all $X^{(p)}, Y^{(p)}$, and assign weight $w_p^{(t)}$ to each sample from population $p$.
    **Fit the Lasso model:**

$$\min_{\beta \in \mathbb{R}^d} \sum_{p=1}^{P} \sum_{i=1}^{N_p} w_p^{(t)} (Y_i^{(p)} - X_i^{(p)^T} \beta)^2 + \lambda_L ||\beta||_1$$

6:     **Compute Population Losses:** Calculate the MSE for the current model $\beta^{(t)}$ on each original, unweighted standardized population $p$:

$$\text{loss}_p^{(t)} = \text{MSE}(\beta^{(t)}; X^{(p)}, Y^{(p)})$$

7:     **Weight Update:** Update population weights for the next iteration:

$$w_p^{(t+1)'} = w_p^{(t)} \exp(\eta \cdot \text{loss}_p^{(t)})$$

    Normalize the weights:

$$w_p^{(t+1)} = \frac{w_p^{(t+1)'}}{\sum_{j=1}^{P} w_j^{(t+1)'}}$$

---

**Distributionally Robust Optimization (DRO) XGBoost**  This method adapts XGBoost to a distributionally robust setting by iteratively re-weighting populations based on worst-case performance. Each population's data is standardized separately. The procedure aims to select the $k$ most important features by accounting for heterogeneity in population-level performance. The method is summarized in Algorithm 3.

---
**Algorithm 3** DRO XGBoost
---

**Initialize** uniform weights $w_p^{(0)} = \frac{1}{P}$ for each population $p = 1, \ldots, P$.
**Standardize** each population's data $(X^{(p)}, Y^{(p)})$ independently.
**for** $t = 1 \ldots T\_\text{max}$ **do**
    **Construct Pooled Dataset with Sample Weights:**
     Form $X_{\text{all}}, Y_{\text{all}}$ by concatenating all $X^{(p)}, Y^{(p)}$, and assign weight $w_p^{(t)}$ to each sample from population $p$.
    **Train XGBoost Model:**
     Fit an XGBoost regressor or classifier (depending on the task) on the pooled data using sample weights.
    **Compute Population Losses:**
     For each population $p$, compute: $\text{loss}_p^{(t)} = \begin{cases} \frac{1}{N_p} \sum_{i=1}^{N_p} (Y_i^{(p)} - \hat{Y}_i^{(p)})^2 & \text{(Regression)} \\ \frac{1}{N_p} \sum_{i=1}^{N_p} \text{logloss}(Y_i^{(p)}, \hat{Y}_i^{(p)}) & \text{(Classification)} \end{cases}$
    **Update Population Weights:** $w_p^{(t+1)'} = w_p^{(t)} \exp(\eta \cdot \text{loss}_p^{(t)}), \quad w_p^{(t+1)} = \frac{w_p^{(t+1)'}}{\sum_{j=1}^{P} w_j^{(t+1)'}}$

---

The $k$ features with the largest feature importance scores from the final XGBoost model are selected.

**Embedded baseline – Embedded MLP**  We include an embedded baseline, namely Embedded MLP which uses an MLP with a learnable feature mask trained via DRO. The MLP has a single hidden layer of size 100. We train the model with the joint objective of MSE minimization (for the

regression task), and L-1 regularization (weighted by hyperparameter $\lambda = 0.01$; we found this value to work best out of $[0.1, 0.01, 0.001]$). The training procedure alternates between neural network training and population weight updates over multiple iterations (with the DRO weights being updated once every 10 epochs of the model training). The learnable mask parameters are constrained to $[0, 1]$ and the top-$k$ features are selected based on the final mask values after training convergence. We train for a maximum of 200 epochs, with early stopping with patience of 10 epochs when the average loss improvement falls below $10^{-4}$.

# D  Results (Continued)

## D.1  Synthetic dataset 1: Linear model

Table 1: Results for synthetic experiment 1 (Section 4.1). Performance comparison across methods and populations for different feature budgets. Total number of features is 15.

| Method | Pop. | Budget 5 | | Budget 10 | |
|---|---|---|---|---|---|
| | | **MLP MSE** $\downarrow$ | **RF MSE** $\downarrow$ | **MLP MSE** $\downarrow$ | **RF MSE** $\downarrow$ |
| **Our Method** | A | **0.2251 $\pm$ 0.0029** | **0.2486 $\pm$ 0.0018** | 0.0342 $\pm$ 0.0004 | 0.0535 $\pm$ 0.0010 |
| | B | 0.2888 $\pm$ 0.0113 | 0.3194 $\pm$ 0.0098 | **0.0207 $\pm$ 0.0005** | **0.0804 $\pm$ 0.0036** |
| | C | 0.2787 $\pm$ 0.0150 | 0.3039 $\pm$ 0.0130 | 0.0039 $\pm$ 0.0004 | 0.0339 $\pm$ 0.0079 |
| DRO Lasso | A | 0.5071 $\pm$ 0.0129 | 0.5434 $\pm$ 0.0191 | **0.0338 $\pm$ 0.0009** | **0.0534 $\pm$ 0.0009** |
| | B | **0.2886 $\pm$ 0.0086** | 0.3194 $\pm$ 0.0026 | 0.0211 $\pm$ 0.0001 | 0.0807 $\pm$ 0.0041 |
| | C | 0.2755 $\pm$ 0.0150 | 0.3031 $\pm$ 0.0113 | 0.0039 $\pm$ 0.0004 | 0.0340 $\pm$ 0.0078 |
| DRO XGB | A | 0.5107 $\pm$ 0.0113 | 0.5441 $\pm$ 0.0188 | 0.0738 $\pm$ 0.0370 | 0.0887 $\pm$ 0.0327 |
| | B | **0.2888 $\pm$ 0.0081** | **0.3187 $\pm$ 0.0030** | 0.0408 $\pm$ 0.0198 | 0.0928 $\pm$ 0.0145 |
| | C | 0.2762 $\pm$ 0.0151 | 0.3032 $\pm$ 0.0112 | **0.0036 $\pm$ 0.0003** | 0.0337 $\pm$ 0.0076 |
| Lasso | A | 1.0087 $\pm$ 0.0153 | 1.0571 $\pm$ 0.0107 | 0.2056 $\pm$ 0.1714 | 0.2106 $\pm$ 0.1625 |
| | B | 0.5744 $\pm$ 0.0051 | 0.6150 $\pm$ 0.0052 | 0.1160 $\pm$ 0.0921 | 0.1460 $\pm$ 0.0763 |
| | C | **0.1080 $\pm$ 0.0107** | **0.1283 $\pm$ 0.0102** | **0.0036 $\pm$ 0.0003** | 0.0340 $\pm$ 0.0076 |
| XGB | A | 1.0136 $\pm$ 0.0076 | 1.0623 $\pm$ 0.0039 | 0.4051 $\pm$ 0.2802 | 0.3905 $\pm$ 0.2579 |
| | B | 0.5702 $\pm$ 0.0008 | 0.6170 $\pm$ 0.0076 | 0.2279 $\pm$ 0.1586 | 0.2373 $\pm$ 0.1307 |
| | C | 0.1314 $\pm$ 0.0291 | 0.1538 $\pm$ 0.0352 | 0.0037 $\pm$ 0.0002 | **0.0338 $\pm$ 0.0077** |
| Embedded MLP | A | 0.7355 $\pm$ 0.2702 | 0.7709 $\pm$ 0.2636 | 0.5336 $\pm$ 0.3503 | 0.5011 $\pm$ 0.3107 |
| | B | 0.5599 $\pm$ 0.1363 | 0.6043 $\pm$ 0.1310 | 0.4170 $\pm$ 0.2951 | 0.4009 $\pm$ 0.2521 |
| | C | 0.2805 $\pm$ 0.1630 | 0.3077 $\pm$ 0.1728 | 0.2801 $\pm$ 0.2117 | 0.2649 $\pm$ 0.1923 |

[1] Values reported as mean $\pm$ standard deviation.
[2] $\downarrow$ indicates lower values are better.
[3] Best results per population metric are highlighted in **bold**.

**Implementation details**  $\alpha$ is initialized values to near 1 by adding random noise to a vector of ones. We use Adam optimzer Kingma [2014] with a learning rate of 0.1. We also use a CosineAnnealing Scheduler for the learning rate, and train the model for 200 epochs. For the kernel estimation, we set the number of nearest neighbours $k = 1000$. We take 10 Monte Carlo samples for estimating the objective. At each epoch, we do a full-batch gradient descent. For the objective, we use the hard-max formulation (setting the SoftMax parameter to `inf`). The penalty term is a reciprocal of the $L_1$ norm of $\alpha$. The entire dataset (combining all splits) is of size 36000. We ran the experiment for 3 different seeds and reported the average over all runs as seen in Figure 1. Experiments were conducted on an Apple MacBook Pro equipped with an Apple M3. We set the budget to 5 and, for comparison, also include the results for an increased budget of 10. The metrics are summarized in Table 1.

**Consistency across seeds**  Our method consistently selects a similar core ordered set of features across (3) different seeds (in decreasing order of importance) [0, 7, 1, 5, 8, 6, 9, 2, 10, 3], [0, 7, 1, 8, 5, 9, 6, 2, 10, 3], [0, 7, 5, 1, 8, 6, 9, 2, 10, 3]. In contrast, baseline methods, especially non-DRO versions, show significant variability, often selecting irrelevant noisy features (e.g., 13, 14) and failing to consistently identify the true signal variables.

## D.2 Synthetic dataset 2: Nonlinear

Table 2: Results for synthetic experiment 1 (Section 4.1). Performance comparison across methods and populations for different feature budgets. Total number of features is 50, and budget is set to 8.

| Method | Pop. | MLP MSE ↓ | RF MSE ↓ |
|---|---|---|---|
| **Our Method** | A | $0.0054 \pm 0.0007$ | $\mathbf{0.0150 \pm 0.0033}$ |
| | B | $\mathbf{0.0062 \pm 0.0010}$ | $\mathbf{0.0205 \pm 0.0016}$ |
| | C | $\mathbf{0.0123 \pm 0.0080}$ | $0.2460 \pm 0.0483$ |
| | D | $0.0114 \pm 0.0015$ | $0.0130 \pm 0.0008$ |
| DRO Lasso | A | $0.4115 \pm 0.0554$ | $0.4020 \pm 0.0560$ |
| | B | $0.4172 \pm 0.0533$ | $0.4230 \pm 0.0511$ |
| | C | $0.3114 \pm 0.2644$ | $0.3816 \pm 0.1018$ |
| | D | $0.3569 \pm 0.0259$ | $0.3375 \pm 0.0089$ |
| DRO XGB | A | $0.0054 \pm 0.0014$ | $0.0150 \pm 0.0034$ |
| | B | $0.0064 \pm 0.0009$ | $0.0206 \pm 0.0017$ |
| | C | $0.0226 \pm 0.0005$ | $\mathbf{0.2460 \pm 0.0479}$ |
| | D | $\mathbf{0.0114 \pm 0.0012}$ | $0.0132 \pm 0.0010$ |
| Lasso | A | $0.0517 \pm 0.0417$ | $0.0616 \pm 0.0433$ |
| | B | $0.0562 \pm 0.0443$ | $0.0657 \pm 0.0390$ |
| | C | $0.4654 \pm 0.0573$ | $0.4581 \pm 0.0676$ |
| | D | $0.0114 \pm 0.0014$ | $0.0130 \pm 0.0012$ |
| XGB | A | $0.0051 \pm 0.0013$ | $0.0151 \pm 0.0033$ |
| | B | $0.0067 \pm 0.0012$ | $0.0206 \pm 0.0015$ |
| | C | $0.0200 \pm 0.0039$ | $0.2471 \pm 0.0475$ |
| | D | $0.0117 \pm 0.0017$ | $0.0131 \pm 0.0007$ |
| Embedded MLP | A | $\mathbf{0.0050 \pm 0.0008}$ | $0.0150 \pm 0.0035$ |
| | B | $0.0062 \pm 0.0014$ | $0.0207 \pm 0.0019$ |
| | C | $0.3030 \pm 0.2435$ | $0.4119 \pm 0.1885$ |
| | D | $0.0119 \pm 0.0019$ | $\mathbf{0.0128 \pm 0.0008}$ |

[1] Values reported as mean $\pm$ standard deviation for Budget 8.
[2] ↓ indicates lower values are better.
[3] Budget refers to the number of features selected.

**Implementation details**  $\boldsymbol{\alpha}$ is initialized values to near 2 by adding random noise to a vector of twos. We use Adam optimzer Kingma [2014] with a learning rate of 0.1. We also use a CosineAnnealing Scheduler for the learning rate, and train the model for 150 epochs. For the kernel estimation, we set the number of nearest neighbours $k = 1000$. We take 50 Monte Carlo samples for estimating the objective. At each epoch, we do a full-batch gradient descent. For the objective, we use the hard-max formulation (setting the SoftMax parameter to `inf`). The penalty term is a reciprocal of the $L_1$ norm of $\boldsymbol{\alpha}$. The entire dataset (combining all splits) is of size $44000$. We ran the experiment for 3 different seeds and reported the average over all runs as seen in Figure 2. Experiments were conducted on an Apple MacBook Pro equipped with an Apple M3. We set the budget to 8. The metrics are summarized in Table 2.

## D.3 ACS Dataset

**Implementation details**  $\boldsymbol{\alpha}$ is intialized values to near 5 by adding random noise to a vector of 5s. We use an Adam optimzer, set initial learning rates of 0.01 with a CosineAnnealing Scheduler for the learning rate, and run the optimization for 120 epochs. For the kernel estimation, we set the number of nearest neighbours $k = 500$. We take 10 Monte Carlo samples for estimating the objective. At each epoch, we do a full-batch gradient descent. For the objective, we use the SoftMax formulation (setting the SoftMax parameter to 10). The penalty term is a reciprocal of the $L_1$ norm of $\boldsymbol{\alpha}$, with $\lambda = 1e - 4$. We ran the experiment for 5 different seeds and reported the average over all runs as seen in Figure 3a. For the ACS dataset, we set the budget to 7 out of 18 features. The metrics are summarized in Table 3. For larger visualizations for greater readability, see Figure 4a.

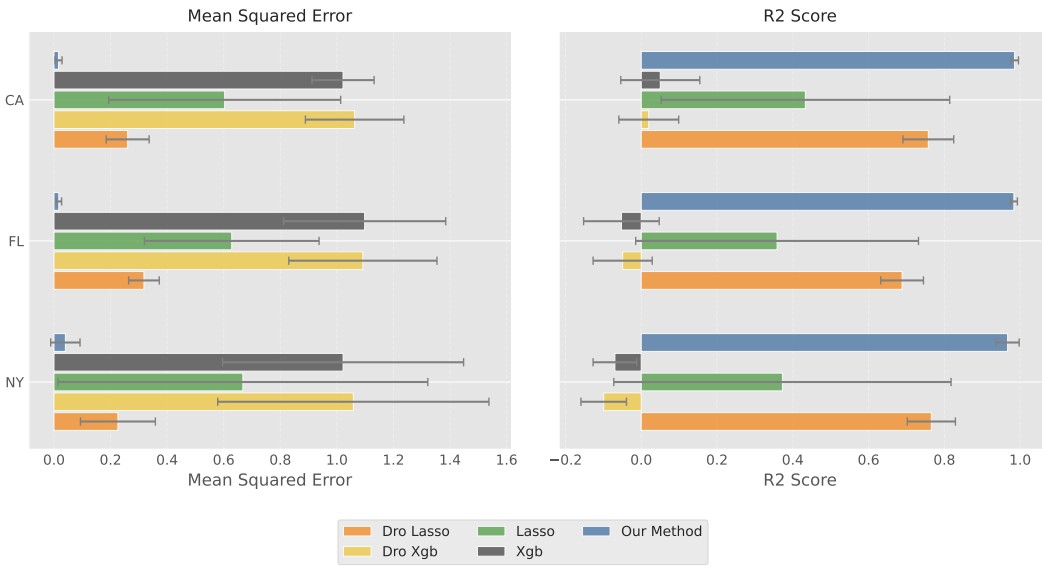

(a) **ACS dataset results**: Comparison of mean squared error (MSE) and $R^2$ score (with standard deviations) for each method across California (CA), Florida (FL), and New York (NY). Our method achieves an order-of-magnitude lower MSE and substantially higher $R^2$ than all baselines, with consistently low run-to-run variance.

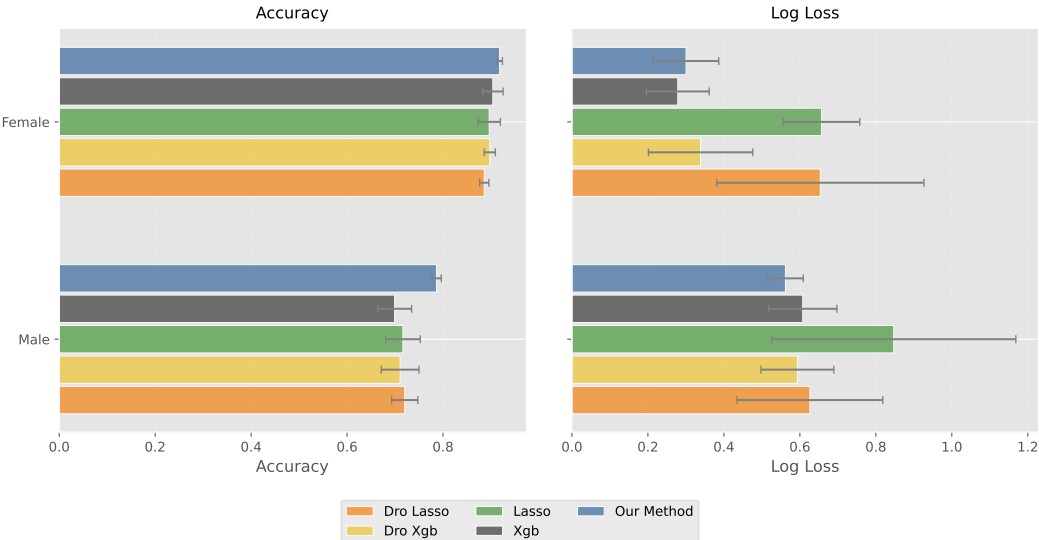

(b) **UCI dataset results**: Downstream classification performance (mean accuracy and log loss with standard deviations) for each method on Female and Male populations. Our method achieves the highest accuracy in both groups while maintaining a comparative log loss and lower run-to-run variance.

Figure 4: **Performance comparison across populations on real datasets using different feature selection methods**.

## D.4 UCI Dataset

$\boldsymbol{\alpha}$ is intialized values to near 5 by adding random noise to a vector of 5s. We use an Adam optimzer, and set initial learning rates to $0.02$ with a CosineAnnealing Scheduler for the learning rate, and run the optimization for 120 epochs. For the kernel estimation, we set the number of nearest neighbours $k = 500$. We take 10 Monte Carlo samples for estimating the objective. At each epoch, we do a full-batch gradient descent. For the objective, we use the SoftMax formulation (setting the SoftMax parameter to 10). The penalty term is a reciprocal of the $L_1$ norm of $\boldsymbol{\alpha}$, with $\lambda = 5e - 6$. Due to computational instability of the Logistic Regression baseline, we ran the experiment for 3 different

Table 3: ACS dataset: Performance comparison across methods and populations

| Method | Population | MSE ↓ | $R^2$ ↑ |
|---|---|---|---|
| Baseline DRO Lasso | CA | 0.260 ± 0.076 | 0.758 ± 0.067 |
| | FL | 0.318 ± 0.054 | 0.688 ± 0.056 |
| | NY | 0.226 ± 0.132 | 0.766 ± 0.064 |
| Baseline DRO XGBoost | CA | 1.063 ± 0.174 | 0.020 ± 0.079 |
| | FL | 1.092 ± 0.262 | -0.049 ± 0.078 |
| | NY | 1.058 ± 0.479 | -0.099 ± 0.060 |
| Baseline Lasso | CA | 0.604 ± 0.410 | 0.433 ± 0.381 |
| | FL | 0.628 ± 0.309 | 0.359 ± 0.373 |
| | NY | 0.668 ± 0.654 | 0.373 ± 0.445 |
| Baseline XGBoost | CA | 1.022 ± 0.110 | 0.050 ± 0.104 |
| | FL | 1.098 ± 0.286 | -0.052 ± 0.100 |
| | NY | 1.022 ± 0.426 | -0.069 ± 0.057 |
| **Our Method** | CA | **0.016 ± 0.012** | **0.985 ± 0.010** |
| | FL | **0.017 ± 0.010** | **0.983 ± 0.009** |
| | NY | **0.040 ± 0.052** | **0.967 ± 0.030** |

[1] Values reported as mean ± standard deviation, rounded to three decimal places.
[2] ↓ indicates lower values are better; ↑ indicates higher values are better.
[3] Best results per population metric are highlighted in **bold**.

Table 4: UCI dataset: Performance comparison across methods and populations

| Method | Population | Accuracy ↑ | Log Loss ↓ |
|---|---|---|---|
| Baseline DRO Lasso | Female | 0.886 ± 0.010 | 0.654 ± 0.273 |
| | Male | 0.720 ± 0.027 | 0.626 ± 0.192 |
| Baseline DRO XGBoost | Female | 0.897 ± 0.012 | 0.339 ± 0.137 |
| | Male | 0.711 ± 0.039 | 0.594 ± 0.096 |
| Baseline Lasso | Female | 0.896 ± 0.023 | 0.657 ± 0.101 |
| | Male | 0.716 ± 0.036 | 0.847 ± 0.321 |
| Baseline XGBoost | Female | 0.904 ± 0.021 | **0.279 ± 0.083** |
| | Male | 0.699 ± 0.035 | 0.608 ± 0.090 |
| **Our Method** | Female | **0.918 ± 0.006** | 0.300 ± 0.086 |
| | Male | **0.786 ± 0.010** | **0.562 ± 0.047** |

[1] Values reported as mean ± standard deviation, rounded to three decimal places.
[2] ↑ indicates higher values are better; ↓ indicates lower values are better.
[3] Best results per population metric are highlighted in **bold**.

seeds and reported the average over all runs as seen in Figure 3b. We set the budget to 5 out of 44 (real and one-hot-encoded) features. The metrics are summarized in Table 4. For larger visualizations for greater readability, see Figure 4b

# E  Additional synthetic experiments

## E.1  Synthetic experiment 1: Linear model with increased dimension

In this section we include additional experiments for the data setting from Section 4.1, **synthetic experiment 1** (linear data setting), when the total number of features is set to 50. Here, the increase in dimension only contributes in additional noise variables, while the meaningful variables are the same as for dimension= 15. We set the budget for feature selection to 5 and, for comparison, also include the results for an increased budget of 10. Figure 5 shows the performance of the downstream

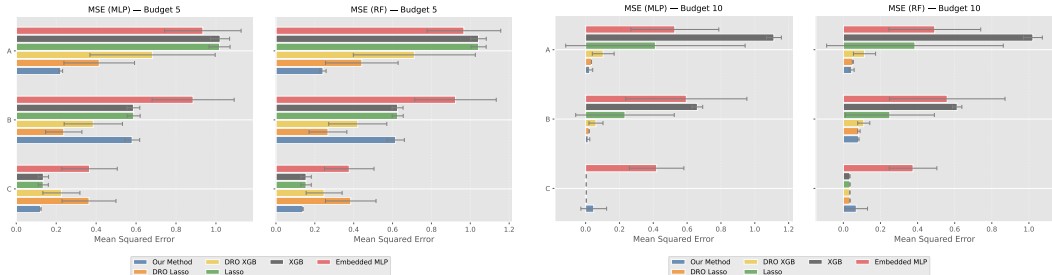

(a) **Budget = 5**: Our method consistently among the top-3 feature selection methods along with DRO-XGBoost and DRO-Lasso, even outperforming both methods in population $C$.

(b) **Budget = 10**: Overall performance improves predictably upon reducing the budget. Our method maintains lower MSE compared to other methods, including DRO-Lasso, and has a balanced performance across populations.

Figure 5: **Performance comparison across populations on synthetic dataset 1 using different feature selection methods**. In each subplot, left: Mean Squared Error of downstream MLP model; right: Mean Squared Error of downstream random forest model. The relative performance of feature selection methods is consistent across the choice of downstream prediction model, with vanilla XGBoost consistently having the worst performance. Here the total number of features is set to 50.

Table 5: Additional results for synthetic experiment 1 (Section 4.1). Performance comparison across methods and populations for different feature budgets. Total number of features is 50.

| Method | Pop. | Budget 5 | | Budget 10 | |
|---|---|---|---|---|---|
| | | MLP MSE ↓ | RF MSE ↓ | MLP MSE ↓ | RF MSE ↓ |
| **Our Method** | A | **0.2227 ± 0.0094** | **0.2420 ± 0.0161** | **0.0241 ± 0.0178** | **0.0450 ± 0.0129** |
| | B | 0.5805 ± 0.0375 | 0.6158 ± 0.0448 | **0.0150 ± 0.0096** | **0.0809 ± 0.0035** |
| | C | **0.1208 ± 0.0034** | **0.1388 ± 0.0011** | 0.0480 ± 0.0761 | 0.0697 ± 0.0605 |
| DRO Lasso | A | 0.4155 ± 0.1769 | 0.4418 ± 0.1866 | 0.0345 ± 0.0004 | 0.0511 ± 0.0029 |
| | B | **0.2377 ± 0.0909** | **0.2677 ± 0.0972** | 0.0206 ± 0.0006 | 0.0835 ± 0.0067 |
| | C | 0.3642 ± 0.1350 | 0.3849 ± 0.1300 | 0.0040 ± 0.0007 | 0.0354 ± 0.0012 |
| DRO XGB | A | 0.6827 ± 0.3136 | 0.7120 ± 0.3138 | 0.1041 ± 0.0645 | 0.1139 ± 0.0597 |
| | B | 0.3857 ± 0.1461 | 0.4209 ± 0.1495 | 0.0613 ± 0.0416 | 0.1098 ± 0.0326 |
| | C | 0.2259 ± 0.0928 | 0.2479 ± 0.0926 | 0.0040 ± 0.0005 | 0.0350 ± 0.0010 |
| Lasso | A | 1.0181 ± 0.0526 | 1.0429 ± 0.0387 | 0.4126 ± 0.5303 | 0.3859 ± 0.4760 |
| | B | 0.5880 ± 0.0323 | 0.6264 ± 0.0284 | 0.2327 ± 0.2912 | 0.2504 ± 0.2401 |
| | C | 0.1351 ± 0.0252 | 0.1550 ± 0.0267 | 0.0037 ± 0.0001 | 0.0354 ± 0.0009 |
| XGB | A | 1.0211 ± 0.0474 | 1.0421 ± 0.0400 | 1.1132 ± 0.0439 | 1.0216 ± 0.0507 |
| | B | 0.5867 ± 0.0319 | 0.6258 ± 0.0282 | 0.6607 ± 0.0314 | 0.6135 ± 0.0252 |
| | C | 0.1356 ± 0.0261 | 0.1551 ± 0.0267 | **0.0036 ± 0.0004** | **0.0347 ± 0.0012** |
| Embedded MLP | A | 0.9346 ± 0.1914 | 0.9670 ± 0.1896 | 0.5275 ± 0.2598 | 0.4928 ± 0.2475 |
| | B | 0.8857 ± 0.2055 | 0.9238 ± 0.2096 | 0.5952 ± 0.3581 | 0.5597 ± 0.3116 |
| | C | 0.3666 ± 0.1390 | 0.3772 ± 0.1274 | 0.4200 ± 0.1610 | 0.3756 ± 0.1282 |

[1] Values reported as mean ± standard deviation.
[2] ↓ indicates lower values are better.
[3] Best results per population metric are highlighted in **bold**.

prediction models on the task of predicting $Y$ for each population, using the features selected using our method and the baselines. The complete metrics can be found in Table 5.

**Results**  Even in the higher dimensional setting, our method's performance is comparable with the best performing baselines (DRO XGBoost and DRO Lasso), while maintaining a balanced performance across populations. Vanilla XGBoost shows the greatest degradation in performance when compared to the lower dimensional setting. Vanilla Lasso and Embedded MLP show the highest

variance across different seeds, while our method, along with DRO Lasso and DROXGBoost has the lowest variance across seeds.

**Implementation details**  $\alpha$ is initialized values to near 1 by adding random noise to a vector of ones. We use Adam optimzer Kingma [2014] with a learning rate of 0.1. We also use a CosineAnnealing Scheduler for the learning rate, and train the model for 150 epochs. For the kernel estimation, we set the number of nearest neighbours $k = 1000$. We take 10 Monte Carlo samples for estimating the objective. At each epoch, we do a full-batch gradient descent. For the objective, we use the hard-max formulation (setting the SoftMax parameter to `inf`). The penalty term is a reciprocal of the $L_1$ norm of $\alpha$. Our entire dataset (combining all splits) is of size 36000. We ran the experiment for 3 different seeds and reported the average over all runs as seen in Figure 5. Experiments were conducted on an Apple MacBook Pro equipped with an Apple M3. We set the budget to 5 and, for comparison, also include the results for an increased budget of 10.

## E.2  Synthetic experiment 3: Sparse Linear

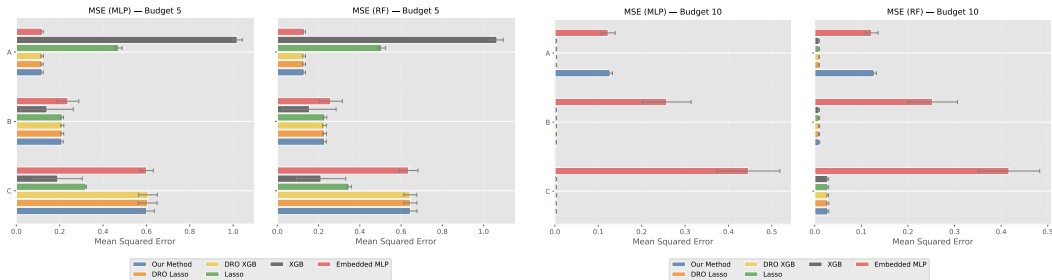

(a) **Budget = 5**: Our method consistently achieves low error across all populations, performing on par with DRO-XGBoost and DRO-Lasso, even outperforming both methods in population $A$.

(b) **Budget = 10**: Overall performance improves predictably upon reducing the budget.

Figure 6: **Performance comparison across populations on synthetic dataset 3 using different feature selection methods**. In each subplot, left: Mean Squared Error of downstream MLP model; right: Mean Squared Error of downstream random forest model. The relative performance of feature selection methods is consistent across the choice of downstream prediction model.

This synthetic dataset comprises three populations $A, B, C$, with 50 features each. The populations have the following proportions in the dataset, and outcome functions:

$$
\begin{aligned}
\text{A (35\%)} \quad & Y = 5X_0 + 4X_{15} + 3X_{30} + \epsilon \\
\text{B (35\%)} \quad & Y = 6X_5 + 5X_{20} + 4X_{35} + \epsilon \\
\text{C (30\%)} \quad & Y = 7X_{10} + 6X_{25} + 5X_{40} + 4X_{45} + \epsilon
\end{aligned}
$$

The noise term follows a base distribution $\epsilon \sim \mathcal{N}(0, 0.1^2)$. Feature correlations are introduced via a generative process where $X_{i+1} = 0.3X_i + 0.7\eta$, with $\eta \sim \mathcal{N}(0, 1)$, inducing structured dependencies across dimensions. The dataset contains 50 features in total, the majority of which are noise. We set the budget for feature selection to 5 and, for comparison, also include the results for an increased budget of 10. Figure 6 shows the performance of the downstream prediction models on the task of predicting $Y$ for each population, using the features selected using our method and the baselines. The complete metrics table can be found in Table 6.

**Results**  We observe that our method performs similarly to the remaining baselines. For budget= 5, all models perform similarly, with the exception of vanilla XGBoost, which has the best performance on populations $B$ and $C$, but seems to do so at the cost of its performance on population $A$. In a group-DRO setting, we would prefer to have a more balanced performance across all populations, as seen in the other methods. For budget= 10, the Embedded MLP performs the worst on all populations. Our method is consistent with the other baselines on populations $B$ and $C$, however, in this setting it performs poorly on population $A$, potentially due to $A$ having the weakest signals.

Table 6: Results for synthetic experiment 3. Performance comparison across methods and populations for different budgets. Total number of features is 50.

| Method | Pop. | Budget 5 | | Budget 10 | |
|---|---|---|---|---|---|
| | | MLP MSE ↓ | RF MSE ↓ | MLP MSE ↓ | RF MSE ↓ |
| **Our Method** | A | $0.1186 \pm 0.0055$ | $0.1299 \pm 0.0064$ | $0.1274 \pm 0.0054$ | $0.1277 \pm 0.0047$ |
| | B | $\mathbf{0.2099 \pm 0.0061}$ | $0.2290 \pm 0.0086$ | $0.0034 \pm 0.0001$ | $0.0093 \pm 0.0014$ |
| | C | $0.6009 \pm 0.0356$ | $0.6453 \pm 0.0318$ | $0.0036 \pm 0.0002$ | $0.0282 \pm 0.0021$ |
| DRO Lasso | A | $\mathbf{0.1179 \pm 0.0053}$ | $\mathbf{0.1296 \pm 0.0066}$ | $0.0038 \pm 0.0003$ | $\mathbf{0.0095 \pm 0.0003}$ |
| | B | $0.2115 \pm 0.0070$ | $0.2295 \pm 0.0088$ | $\mathbf{0.0033 \pm 0.0004}$ | $0.0092 \pm 0.0013$ |
| | C | $0.6054 \pm 0.0438$ | $0.6456 \pm 0.0310$ | $\mathbf{0.0034 \pm 0.0001}$ | $0.0281 \pm 0.0021$ |
| DRO XGB | A | $0.1189 \pm 0.0057$ | $0.1298 \pm 0.0066$ | $0.0038 \pm 0.0000$ | $\mathbf{0.0095 \pm 0.0004}$ |
| | B | $0.2118 \pm 0.0071$ | $0.2286 \pm 0.0092$ | $0.0035 \pm 0.0003$ | $0.0092 \pm 0.0014$ |
| | C | $0.6073 \pm 0.0432$ | $0.6447 \pm 0.0311$ | $0.0037 \pm 0.0004$ | $\mathbf{0.0280 \pm 0.0018}$ |
| Lasso | A | $0.4729 \pm 0.0153$ | $0.5066 \pm 0.0185$ | $0.0039 \pm 0.0004$ | $\mathbf{0.0095 \pm 0.0004}$ |
| | B | $0.2115 \pm 0.0057$ | $0.2309 \pm 0.0090$ | $0.0036 \pm 0.0004$ | $0.0092 \pm 0.0013$ |
| | C | $0.3202 \pm 0.0029$ | $0.3485 \pm 0.0106$ | $0.0037 \pm 0.0001$ | $0.0281 \pm 0.0018$ |
| XGB | A | $1.0188 \pm 0.0235$ | $1.0651 \pm 0.0303$ | $\mathbf{0.0036 \pm 0.0003}$ | $\mathbf{0.0095 \pm 0.0004}$ |
| | B | $0.1414 \pm 0.1214$ | $\mathbf{0.1564 \pm 0.1285}$ | $0.0034 \pm 0.0002$ | $\mathbf{0.0091 \pm 0.0013}$ |
| | C | $\mathbf{0.1915 \pm 0.1125}$ | $\mathbf{0.2121 \pm 0.1196}$ | $0.0037 \pm 0.0004$ | $0.0280 \pm 0.0019$ |
| Embedded MLP | A | $0.1188 \pm 0.0051$ | $0.1306 \pm 0.0054$ | $0.1229 \pm 0.0163$ | $0.1222 \pm 0.0138$ |
| | B | $0.2381 \pm 0.0499$ | $0.2595 \pm 0.0562$ | $0.2579 \pm 0.0562$ | $0.2530 \pm 0.0534$ |
| | C | $0.6004 \pm 0.0314$ | $0.6364 \pm 0.0459$ | $0.4461 \pm 0.0724$ | $0.4168 \pm 0.0659$ |

[1] Values reported as mean ± standard deviation.
[2] ↓ indicates lower values are better.
[3] Budget refers to the number of features selected.

**Implementation details**   $\alpha$ is initialized values to near 2 by adding random noise to a vector of twos. We use Adam optimzer Kingma [2014] with a learning rate of 0.1. We also use a CosineAnnealing Scheduler for the learning rate, and train the model for 150 epochs. For kernel estimation, we set the number of nearest neighbors $k = 1000$. We take 50 Monte Carlo samples for estimating the objective. At each epoch, we do a full-batch gradient descent. For the objective, we use the hard-max formulation (setting the SoftMax parameter to `inf`). The penalty term is a reciprocal of the $L_1$ norm of $\alpha$. Our entire dataset (combining all splits) is of size 36000. We ran the experiment for 3 different seeds and reported the average over all runs as seen in Figure 1. Experiments were conducted on an Apple MacBook Pro equipped with an Apple M3. We set the budget to 5 and, for comparison, also include the results for an increased budget of 10.

