# OpenReview forum: "Distributionally Robust Feature Selection"
_NeurIPS.cc/2025/Conference — NeurIPS 2025 poster_

### Official Review · Reviewer_789K · 2025-06-23

**Clarity:** 3
**Significance:** 2
**Originality:** 3
**Rating:** 4
**Confidence:** 4

**Summary:**

The paper under review proposes a new method for distributional robust feature selection, using an insightful continuous relaxation of an intractable, non-convex problem. The authors provide helpful intuition for their method and evaluate it via simulations and real data analysis.

**Questions:**

1. The form of the relaxation of problem (1) as detailed in Section 3 is tailored for a square loss. However, the authors use their method in the experimentation with the cross entropy loss to satisfying effect. What is the authors' understanding of the validity of their proposed procedure under different loss functions or distributions on the outcome $Y$? How does the performance of the proposed method change under different, perhaps non-smooth loss functions?

2. How does the proposed method fare under larger dimensions? The simulations are run with only 14 features, and the real data examples are not much larger. Is there a severe computational burden in higher dimensions? It would be very helpful to see a table of runtime statistics, since this is nominally an optimization paper.

3. How does the method perform under different data generating mechanisms between the train and test data? More diverse simulations would be extremely valuable to this paper.

**Ethical Concerns:**

["NO or VERY MINOR ethics concerns only"]

**Final Justification:**

In the rebuttal period, the authors provided strong simulation results in response to my questions, as well as the points raised by the other reviewers. I think that these simulation results, which cover a wider range of scenarios than those explored in the original version of the paper, improve the evidence of the efficacy of the proposed method. I still hold minor reservations over the fact that the proposed algorithm is derived using the bias-variance decomposition of the MSE, which is why I keep my score at 4 rather than raise it higher. However, I understand that this would constitute a non-trivial extension to the submitted paper and is not feasible to address within the rebuttal period.

**Limitations:**

yes

**Quality:**

2

**Strengths And Weaknesses:**

The primary strength of the paper is its intuitive derivation of the method. Section 3 (Methods) is well written and clear. Furthermore, the authors present a very early (if not explicitly the first) attempt at a solution for distributionally robust feature selection. Since distributionally robust methods are growing in popularity, this constitutes a valuable contribution to the machine learning field.

However, the evaluation of the proposed method leaves much to be desired. The authors only evaluate the methods under one data generating mechanism, with the test set being drawn using the same functional relationship between covariates and outcome as the test data. The advantage of DRO in principle is that good performance is guaranteed for nearly arbitrary distribution shifts - this is not explored at all in the simulations. I would like to see a more expansive suite of simulations exploring different functions forms between the covariates and outcome, as well as varying degrees of distribution shift between the training and test datasets. Since the authors do not provide any theoretical guarantees for their proposed method, I would like to see more empirical evidence for the effectiveness of their method in practice.

---

> ### Author Rebuttal · Authors · 2025-07-30
>
> We thank reviewer 789K for their feedback and suggestions, which we address below.
>
> ## Questions:
> > Q1: *The form of the relaxation of problem (1) as detailed in Section 3 is tailored for a square loss. However, the authors use their method in the experimentation with the cross entropy loss to satisfying effect. What is the authors' understanding of the validity of their proposed procedure under different loss functions or distributions on the outcome? How does the performance of the proposed method change under different, perhaps non-smooth loss functions?*
>
> Our derivation uses the bias-variance decomposition specific to MSE. However, there exist generalized variance decompositions for other loss functions. Using such generalized decompositions could allow our method to directly optimize cross-entropy or other losses, but this is a nontrivial extension we leave for future work. Our empirical results with cross-entropy (UCI experiment) indeed suggest that optimizing MSE in our method may transfer well to other losses, particularly losses that are proper scoring rules akin to MSE.
>
> > Q2: *How does the proposed method fare under larger dimensions? The simulations are run with only 14 features, and the real data examples are not much larger. Is there a severe computational burden in higher dimensions? It would be very helpful to see a table of runtime statistics, since this is nominally an optimization paper.*
>
> Our method requires $O(P\cdot b\cdot n\cdot K\cdot d)$ operations per iteration, where $P$ is the number of populations, $b$ is the number of Monte Carlo samples, $n =\max_p n_p$ is the maximum population size, $K$ is the number of nearest neighbors used for kernel weight computation, and feature dimensionality $d$.
>
> We now include a synthetic experiment where we vary the dimension of the data under the same generative process (increasing the number or irrelevant/noise variables). Pl. see the last section of our rebuttal for the experiment results.
> The runtime statistics per epoch (averaged across 3 runs) for this experiment are given in the table below:
> |Dimension (d)|Time per epoch (s) (Mean±Std)|
> |---|---|
> |15|1.8988±0.0026|
> |50|1.9217±0.0044|
>
> For $d=15$, we run 50-64 epochs, while for $d=50$, we run 126-150 epochs.
>
> > Q3: *How does the method perform under different data generating mechanisms between the train and test data? More diverse simulations would be extremely valuable to this paper.*
>
> We wish to clarify that our method is designed for Group DRO, which we discuss in our related work section (Section 1.1, citing Sagawa et al. [2019]). Group-DRO optimizes for worst-case performance across a priori known subpopulations (e.g., demographic groups, different hospitals). This setting is distinct from DRO over arbitrary distribution shifts. Thus our experimental setup uses predefined groups rather than exploring arbitrary functional form changes.
>
> We have run new synthetic experiments with different generative processes such as a purely linear model, a mixed linear-nonlinear model with heterogeneous noise, and a sparse linear model with correlated noise variables. Across these, our method outperforms the baselines or closely follows the best performing ones. We include only the first experiment in the last section of this rebuttal due to character limits. Our final paper will have more experiments.
>
> ## Additional comments
>
> > C1: *The test set being drawn using the same functional relationship between covariates and outcome as the test data. The advantage of DRO in principle is that good performance is guaranteed for nearly arbitrary distribution shifts - this is not explored at all in the simulations. I would like to see a more expansive suite of simulations exploring different functions forms between the covariates and outcome, as well as varying degrees of distribution shift between the training and test datasets. Since the authors do not provide any theoretical guarantees for their proposed method, I would like to see more empirical evidence for the effectiveness of their method in practice.*
>
> We request the reviewer to refer to our response to Q3. Additionally, our algorithmic framework can be extended to more general f-divergence DRO. This would involve fitting a series of conditional mean models for the worst-case distribution at each optimization step, rather than one model per predefined group. The core procedure remains the same, highlighting the flexibility of our approach. We agree that exploring these broader distribution shift scenarios is a valuable direction for future work.
>
> Wrt more simulations, we have run new experiments on synthetic datasets. We include only one of them here due to character limits. However, you can find the other experiments in our response to reviewer ggn5. They also include:
> - Additional downstream model for evaluating performance: we now include an MLP, along with the existing random forest (RF), as a downstream task model.
> - New baseline methods: we've added Forward Selection and Backward Elimination, with their DRO variants, (along with the existing baseline methods in the original paper).
> - Additional data generation settings to explore different covariate-outcome functional relationships.
>
> # Experiment
> **Population Structure**:
> - A (40%): $$ Y = 8X_0 + 6X_1 - 4X_2 + 3X_3 + 2X_4 + \epsilon $$
> - B (35%): $$ Y = -8X_0 - 6X_1 + 4X_2 - 3X_3 - 2X_4 + 8X_5 + 6X_6 + \epsilon $$
> - C (25%): $$ Y = 10X_7 + 8X_8 + 6X_9 - 5X_{10} + \epsilon $$
> $$\text{Noise:} \quad \epsilon \sim \mathcal{N}(0, 0.1^2)$$
>
> We use a dataset of size 36000, with a 40-60 train-(test/val) split for each population. We report the results for two feature budgets (5 and 10).
>
> |Dimension|Method|Population|MSE±Std (MLP) — Budget 5|MSE±Std (RF) — Budget 5|MSE±Std (MLP) — Budget 10|MSE±Std (RF) — Budget 10|
> |---|---|---|---|---|---|---|
> |15|Our Method|A|0.2251±0.0029|0.2486±0.0018|0.0342±0.0004|0.0535±0.0010|
> |||B|0.2888±0.0113|0.3194±0.0098|0.0207±0.0005|0.0804±0.0036|
> |||C|0.2787±0.0150|0.3039±0.0130|0.0039±0.0004|0.0339±0.0079|
> |50||A|0.2227±0.0094|0.2420±0.0161|0.0241±0.0178|0.0450±0.0129|
> |||B|0.5805±0.0375|0.6158±0.0448|0.0150±0.0096|0.0809±0.0035|
> |||C|0.1208±0.0034|0.1388±0.0011|0.0480±0.0761|0.0697±0.0605|
> |15|(BL) DRO Lasso|A|0.5071±0.0129|0.5434±0.0191|0.0338±0.0009|0.0534±0.0009|
> |||B|0.2886±0.0086|0.3194±0.0026|0.0211±0.0001|0.0807±0.0041|
> |||C|0.2755±0.0150|0.3031±0.0113|0.0039±0.0004|0.0340±0.0078|
> |50||A|0.4155±0.1769|0.4418±0.1866|0.0345±0.0004|0.0511±0.0029|
> |||B|0.2377±0.0909|0.2677±0.0972|0.0206±0.0006|0.0835±0.0067|
> |||C|0.3642±0.1350|0.3849±0.1300|0.0040±0.0007|0.0354±0.0012|
> |15|(BL) DRO XGB|A|0.5107±0.0113|0.5441±0.0188|0.0738±0.0370|0.0887±0.0327|
> |||B|0.2888±0.0081|0.3187±0.0030|0.0408±0.0198|0.0928±0.0145|
> |||C|0.2762±0.0151|0.3032±0.0112|0.0036±0.0003|0.0337±0.0076|
> |50||A|0.6827±0.3136|0.7120±0.3138|0.1041±0.0645|0.1139±0.0597|
> |||B|0.3857±0.1461|0.4209±0.1495|0.0613±0.0416|0.1098±0.0326|
> |||C|0.2259±0.0928|0.2479±0.0926|0.0040±0.0005|0.0350±0.0010|
> |15|(BL) Lasso|A|1.0087±0.0153|1.0571±0.0107|0.2056±0.1714|0.2106±0.1625|
> |||B|0.5744±0.0051|0.6150±0.0052|0.1160±0.0921|0.1460±0.0763|
> |||C|0.1080±0.0107|0.1283±0.0102|0.0036±0.0003|0.0340±0.0076|
> |50||A|1.0181±0.0526|1.0429±0.0387|0.4126±0.5303|0.3859±0.4760|
> |||B|0.5880±0.0323|0.6264±0.0284|0.2327±0.2912|0.2504±0.2401|
> |||C|0.1351±0.0252|0.1550±0.0267|0.0037±0.0001|0.0354±0.0009|
> |15|(BL) XGB|A|1.0136±0.0076|1.0623±0.0039|0.4051±0.2802|0.3905±0.2579|
> |||B|0.5702±0.0008|0.6170±0.0076|0.2279±0.1586|0.2373±0.1307|
> |||C|0.1314±0.0291|0.1538±0.0352|0.0037±0.0002|0.0338±0.0077|
> |50||A|1.0211±0.0474|1.0421±0.0400|1.1132±0.0439|1.0216±0.0507|
> |||B|0.5867±0.0319|0.6258±0.0282|0.6607±0.0314|0.6135±0.0252|
> |||C|0.1356±0.0261|0.1551±0.0267|0.0036±0.0004|0.0347±0.0012|
> |15|(BL) DRO Forward Selection|A|0.2257±0.0030|0.2498±0.0005|0.0344±0.0009|0.0534±0.0012|
> |||B|0.3274±0.0747|0.3596±0.0808|0.0208±0.0010|0.0806±0.0041|
> |||C|0.3752±0.1545|0.3991±0.1581|0.0037±0.0002|0.0341±0.0079|
> |50||A|0.3735±0.1506|0.3996±0.1636|0.0341±0.0006|0.0514±0.0029|
> |||B|0.3729±0.0553|0.3995±0.0553|0.0209±0.0012|0.0833±0.0066|
> |||C|0.4735±0.0805|0.5124±0.0883|0.0038±0.0001|0.0354±0.0012|
> |15|(BL) DRO Backward Elimination|A|0.0314±0.0008|0.0492±0.0008|0.0341±0.0007|0.0535±0.0010|
> |||B|0.1833±0.0047|0.2129±0.0066|0.0209±0.0006|0.0806±0.0037|
> |||C|1.0281±0.0344|1.0455±0.0548|0.0038±0.0002|0.0342±0.0079|
> |50||A|0.0320±0.0012|0.0474±0.0024|0.0346±0.0005|0.0512±0.0030|
> |||B|0.1757±0.0093|0.2079±0.0149|0.0208±0.0008|0.0834±0.0068|
> |||C|1.0008±0.0669|1.0355±0.0635|0.0038±0.0003|0.0354±0.0012|
> |15|(BL) Forward Selection|A|1.0183±0.0147|1.0430±0.0118|1.0887±0.0176|1.0132±0.0219|
> |||B|0.5745±0.0043|0.6221±0.0061|0.6430±0.0226|0.5956±0.0086|
> |||C|1.0298±0.0587|1.0571±0.0500|0.3572±0.3066|0.3407±0.2694|
> |50||A|1.0132±0.0372|1.0408±0.0435|1.1209±0.0483|1.0224±0.0403|
> |||B|0.5879±0.0331|0.6276±0.0384|0.6509±0.0360|0.6109±0.0362|
> |||C|0.9808±0.1161|1.0040±0.0989|1.1410±0.1413|0.9882±0.1095|
> |15|(BL) Backward Elimination|A|0.9972±0.0199|1.0427±0.0126|1.1153±0.0405|1.0136±0.0219|
> |||B|0.5672±0.0081|0.6077±0.0068|0.6486±0.0288|0.5954±0.0068|
> |||C|0.4222±0.2690|0.4523±0.2803|0.3514±0.3013|0.3399±0.2682|
> |50||A|1.0095±0.0393|1.0463±0.0447|1.1267±0.0415|1.0244±0.0405|
> |||B|0.5897±0.0341|0.6267±0.0300|0.6640±0.0208|0.6111±0.0364|
> |||C|0.9883±0.1042|1.0071±0.0912|1.1047±0.1156|0.9895±0.1071|
> ---
>
> ## Discussion
> Although the generative process here is linear and variables $X_0$ to $X_4$ have strong effects in both A and B, the signs of their coefficients are reversed between populations. This reduces the effectiveness of LASSO, which tends to select features based on average effects across all data. As a result, vanilla LASSO does not achieve the best performance even in this linear setting.
>
> For budget=$5$, our method outperforms most baselines, and has a balanced performance across populations. For budget=$10$, our method is comparable with the best performing baselines.

---

> > ### Comment · Reviewer_789K · 2025-08-04
> >
> > I thank the authors for providing answers to my questions and additional simulation details. Also, thank you for clarifying that your framework corresponds to Group-DRO, rather than DRO over arbitrary distribution shifts. I think that the paper will greatly benefit from the additional simulations that the authors have run in response to my questions as well as the questions of the other reviewers. I will increase my score to a 4 accordingly.
> >
> > I think that it would be valuable to include a brief remark in your paper regarding the bias-variance decomposition with other loss functions, even if you do not perform the full derivation in this work.

---

### Official Review · Reviewer_Guhq · 2025-07-01

**Clarity:** 3
**Significance:** 3
**Originality:** 3
**Rating:** 4
**Confidence:** 3

**Summary:**

The paper introduces Distributionally Robust Feature Selection (DRO-FS), a feature selection method aiming to perform well simultaneously across multiple sub-populations of the data. Instead of using masks, the authors present an interesting mechanism for feature selection: injecting white Gaussian noise into the features. An informative feature would be injected with a small variance noise, whereas a non-informative feature would be injected with high variance noise. Instead of differentiating through an inner training loop, the authors derive an objective based on the conditional variance of the Bayes-optimal predictor, which depends on the variance of the noise through a closed-form kernel weight expression. They use SGD to optimize the worst-case loss across groups. The proposed method is demonstrated using synthetic data and two real-world datasets for predicting household income. The proposed approach yields lower worst-group error than Lasso, XGBoost, and the Group-DRO variants, sometimes by an order of magnitude.

**Questions:**

1. Writing - please revise the writing and the derivation of the mathematical formulation, add a concise algorithm box and/or a block diagram of the proposed method to improve clarity.

2. Discuss other methods for feature selection and compare their performance to the proposed approach's.

3. Add other data sets AND and other downstream predictors.

4. Interpretability - report the selected features and compare to other methods of feature selection to demonstrate the advantage of the proposed method.

**Ethical Concerns:**

["NO or VERY MINOR ethics concerns only"]

**Final Justification:**

The authors have fully addressed all of my comments. Together with their responses to the other reviewers’ remarks, the authors resolved every concern raised during the review process. In light of these revisions, I am updating my score to 4.

**Limitations:**

yes

**Quality:**

2

**Strengths And Weaknesses:**

Strengths

* Previous methods are optimized for a specific model, not for a subpopulation, meaning that they ignore distribution shifts.
* The noise injection, along with the Bayes-risk objective, is rather cool. The method is model-agnostic, and there's no need for backpropagating through training.
* The derivation seems correct.
* The authors provide anonymized code
* Training details, architecture choices, and optimization schedules are neatly provided.

Weaknesses

* While the code idea of the paper is interesting, it is poorly presented. The formulation is cumbersome, lacks clarity, and is hard to follow. There are no clear building blocks. Just a steady stream of claims. I think the authors should reformulate the way they derive the equations, use definitions, lemmas, and theorems, and provide proofs. Even if that would end up taking more space, the proofs could go to the appendix. Still, I think it would make the reading more fluent and add clarity to the paper.

* A concise algorithm box and/or a block diagram of the proposed method would improve clarity.

* How does runtime scale with d (features) and n (samples)?
* After learning continuous α, how is the final subset obtained? A hard top-k cut or magnitude-based sorting? I did not find an explicit description. The authors should provide a full description, as this is a fundamental issue, and discuss stability across random seeds.
* So many other methods for feature selection since lasso were not mentioned at all!
* The authors claim the proposed method is model agnostic, yet they demonstrate the performance of their method using only one choice of downstream model (Random Forest). To convince their choice of features is indeed robust, they should use additional downstream models that are not tree-based.
* The synthetic task is highly nonlinear. Including a linear-target setting would indicate when Lasso-type methods might still be sufficient.
* Report the learned sparsity pattern overlap across different random seeds to quantify selection stability.

* Line 133 >=0 at the end is a sub-index of R?
* Line 157 – incurr should be incur
* Line 205 "with with"
* Line 2015 "involves requires"
* Line 222 "By applying .." has no subject
* Line 443 – "initialized"
* Line 452 a missing reference -  ??
* "Moreoever"  should be Moreover appears twice

---

> ### Author Rebuttal · Authors · 2025-07-30
>
> We thank reviewer Guhq for their feedback and suggestions, which we address below.
> ## Questions:
> > Q1: *Writing-please revise the writing and the derivation of the mathematical formulation, add a concise algorithm box and/or a block diagram of the proposed method to improve clarity.*
>
> We thank the reviewer for their suggestions on improving Section 3. We have added an algorithm box summarizing our method in the appendix for greater clarity. We also incorporate the suggestion to present our main results as follows:
>
> **Theorem 1 (Population-Level Objective)**
> Under the noise-based relaxation $S(\alpha) = X + \epsilon(\alpha)$ where $\epsilon(\alpha) \sim \mathcal{N}(0, \text{diag}(\alpha))$, the distributionally robust feature selection problem:
>
> $$\min_{\alpha} \max_{P_i \in \mathcal{P}} \mathbb{E}_{S(\alpha),Y \sim P_i}[\mathcal{L}(Y, f_i^*(S(\alpha)))]+\lambda \text{Reg}(\alpha)$$
>
> where $f_i^*(s)=\mathbb{E}[Y|S(\alpha)=s, P_i]$ is the Bayes-optimal predictor, is equivalent to:
>
> $$\min_{\alpha} \max_{P_i \in \mathcal{P}} ( -\mathbb{E}_{S(\alpha) \sim P_i}[\mathbb{E}[\mu_i(X)|S(\alpha)]^2] )+\lambda \mathrm{Reg}(\alpha)$$
>
> where $\mu_{i}(X)=\mathbb{E}[Y|X, P_i]$.
>
> ---
> Given samples $\{(X_i^j\, Y_i^j)\}_{j=1}^{n_i}$ , from each population $P_i$, let $\hat{\mu}_i(X)$ be an estimator of $\mu_i(X) = \mathbb{E}[Y|X]$ trained on these samples. The empirical form of the population objective from Theorem 1 is:
>
> $$\min_{\alpha} \max_{P_i \in \mathcal{P}} \left( -\hat{\mathbb{E}}_{S(\alpha)}\left[\hat{\mathbb{E}}[\hat{\mu}_i(X)|S(\alpha)]^2\right] \right)+\lambda \text{Reg}(\alpha)$$
>
> where the expectation $\hat{\mathbb{E}}[\hat{\mu_i}(X)|S(\alpha)]$ is taken with respect to the empirical distribution $\hat{P_i}(X)=\frac{1}{n_i}\sum_{j=1}^{n_i} \delta_{X_i^j}(X)$.
>
> **Theorem 2 (Kernel Form Equivalence)**
>
> The empirical expectation
>
> $$\hat{\mathbb{E}}_{S(\alpha)}\left[\hat{\mathbb{E}}[\hat{\mu_i}(X)|S(\alpha)]^2\right]$$
>
> is equivalent to:
>
> $$\hat{\mathbb{E}}\_{S(\alpha)}[(\sum_{j=1}^{n_i} w_i^j (S(\alpha),\alpha) \hat{\mu}_i (X_i^j))^2]$$
>
> where the weights are:
>
> $$w_i^j(S(\alpha), \alpha)=\frac{\exp(\{-\frac{1}{2}(X_i^j-S(\alpha))^T \text{diag}(\alpha)^{-1}(X_i^j-S(\alpha))\})}{\sum_{k=1}^{n_i} \exp(\{-\frac{1}{2}(X_i^k-S(\alpha))^T \text{diag}(\alpha)^{-1}(X_i^k-S(\alpha))\})}$$
>
> ---
>
> The proofs of these results are exactly the step-by-step derivations currently presented in Section 3. We will create a section in the appendix where we expand the proofs further to improve clarity.
>
> > Q2: *Discuss other methods for feature selection and compare their performance to the proposed approach's.*
>
> We now include Forward Selection (FS) and Backward Elimination (BE) (+ their DRO variants) as baselines in our experiments.
> The implementations use k-fold cross-validation to score feature subsets. FS iteratively adds the feature that most improves the cross-validation score. BE starts with all features and iteratively removes the one whose absence hurts performance the least.
> The DRO variants also follow this greedy logic but optimize for robustness- evaluating feature subsets by finding the worst-case cross-validation score across all individual data populations.
> We have added experimental results on a new synthetic dataset that includes these baselines to this rebuttal. Our method generally outperforms these baselines across the evaluated metric.
>
> > Q3: *Add other data sets AND and other downstream predictors.*
>
> We now include an MLP (single hidden layer of size 100) as a downstream model in our evaluations, and will do so for all the experiments in the final paper. We have conducted new synthetic experiments-we include only one of them here due to character limits, however our final paper will include more experiments. Pl. see the last section of our rebuttal for the new experiment.
>
> > Q4: *Interpretability-report the selected features and compare to other methods of feature selection to demonstrate the advantage of the proposed method.*
>
> We will add the selected features for each run in the appendix. Our method is stable across seeds in the subsets of variables that it selects and the order in which it selects them, differing only in a few values. Pl. see the last section of our rebuttal for an example.
>
> ## Additional comments
> > C1: *How does runtime scale with d (features) and n (samples)?*
>
> Our method requires $O(P \cdot b \cdot n \cdot K \cdot d)$ operations per iteration, where $P$ is the number of populations, $b$ is the number of Monte Carlo samples, $n=\max_p n_p$ is the maximum population size, $K$ is the number of nearest neighbors used for kernel weight computation, and $d$ is the feature dimensionality.
>
> > C2: *After learning continuous α, how is the final subset obtained?*
>
> The final subset is obtained by selecting those variables $i$ with the smallest $\alpha_{i}$ values (corresponding to minimum noise). We will explicitly state this in our paper.
>
> > C3: *Experiments: Use additional downstream models that are not tree-based, Including a linear-target setting would indicate when Lasso-type methods might still be sufficient. Report the learned sparsity pattern overlap across different random seeds to quantify selection stability.*
>
> Our new experiments include:
> - Additional downstream model for evaluating performance: we now include an MLP (single hidden layer of size 100), along with the existing random forest (RF), as a downstream task model.
> - New baseline methods: we've added Forward Selection and Backward Elimination, with their DRO variants
> - Additional data generation settings to explore different covariate-outcome functional relationships. Three of these are included in our response to reviewer ggn5: a purely linear model, a mixed linear-nonlinear model with heterogeneous noise, and a sparse linear model with correlated noise variables. Our method outperforms the baselines or closely follows the best performing ones. We include only the first here due to character limits.
>
> > C4: *Typos*
>
> We thank the reviewer for bringing these typos to our notice, we have corrected them. Wrt ``Line 133 >=0 at the end is a sub-index of R?``, we use this notation for $m$ dimensional vectors whose entries are non-negative real numbers.
>
> # New Synthetic Experiment: Linear generative process
> We set dimension $d=15$.
>
> **Population Structure**:
> - A (40%): $$ Y = 8X_0 + 6X_1 - 4X_2 + 3X_3 + 2X_4 + \epsilon $$
> - B (35%): $$ Y = -8X_0 - 6X_1 + 4X_2 - 3X_3 - 2X_4 + 8X_5 + 6X_6 + \epsilon $$
> - C (25%): $$ Y = 10X_7 + 8X_8 + 6X_9 - 5X_{10} + \epsilon $$
> $$\text{Noise:} \quad \epsilon \sim \mathcal{N}(0, 0.1^2)$$
>
> The train-test-val sets for Pops A, B, andChad 14400, 12600, and 9000 samples respectively, and we used a 40-60 train-(test+val) split for each population.
>
> |Method|Population|MSE±Std (MLP) — Budget 5|MSE±Std (RF) — Budget 5|MSE±Std (MLP) — Budget 10|MSE±Std (RF) — Budget 10|
> |---|---|---|---|---|---|
> |Our Method|A|0.2251±0.0029|0.2486±0.0018|0.0342±0.0004|0.0535±0.0010|
> ||B|0.2888±0.0113|0.3194±0.0098|0.0207±0.0005|0.0804±0.0036|
> ||C|0.2787±0.0150|0.3039±0.0130|0.0039±0.0004|0.0339±0.0079|
> |(BL) DRO Lasso|A|0.5071±0.0129|0.5434±0.0191|0.0338±0.0009|0.0534±0.0009|
> ||B|0.2886±0.0086|0.3194±0.0026|0.0211±0.0001|0.0807±0.0041|
> ||C|0.2755±0.0150|0.3031±0.0113|0.0039±0.0004|0.0340±0.0078|
> |(BL) DRO XGB|A|0.5107±0.0113|0.5441±0.0188|0.0738±0.0370|0.0887±0.0327|
> ||B|0.2888±0.0081|0.3187±0.0030|0.0408±0.0198|0.0928±0.0145|
> ||C|0.2762±0.0151|0.3032±0.0112|0.0036±0.0003|0.0337±0.0076|
> |(BL) Lasso|A|1.0087±0.0153|1.0571±0.0107|0.2056±0.1714|0.2106±0.1625|
> ||B|0.5744±0.0051|0.6150±0.0052|0.1160±0.0921|0.1460±0.0763|
> ||C|0.1080±0.0107|0.1283±0.0102|0.0036±0.0003|0.0340±0.0076|
> |(BL) XGB|A|1.0136±0.0076|1.0623±0.0039|0.4051±0.2802|0.3905±0.2579|
> ||B|0.5702±0.0008|0.6170±0.0076|0.2279±0.1586|0.2373±0.1307|
> ||C|0.1314±0.0291|0.1538±0.0352|0.0037±0.0002|0.0338±0.0077|
> |(BL) DRO Forward Selection|A|0.2257±0.0030|0.2498±0.0005|0.0344±0.0009|0.0534±0.0012|
> ||B|0.3274±0.0747|0.3596±0.0808|0.0208±0.0010|0.0806±0.0041|
> ||C|0.3752±0.1545|0.3991±0.1581|0.0037±0.0002|0.0341±0.0079|
> |(BL) DRO Backward Elimination|A|0.0314±0.0008|0.0492±0.0008|0.0341±0.0007|0.0535±0.0010|
> ||B|0.1833±0.0047|0.2129±0.0066|0.0209±0.0006|0.0806±0.0037|
> ||C|1.0281±0.0344|1.0455±0.0548|0.0038±0.0002|0.0342±0.0079|
> |(BL) Forward Selection|A|1.0183±0.0147|1.0430±0.0118|1.0887±0.0176|1.0132±0.0219|
> ||B|0.5745±0.0043|0.6221±0.0061|0.6430±0.0226|0.5956±0.0086|
> ||C|1.0298±0.0587|1.0571±0.0500|0.3572±0.3066|0.3407±0.2694|
> |(BL) Backward Elimination|A|0.9972±0.0199|1.0427±0.0126|1.1153±0.0405|1.0136±0.0219|
> ||B|0.5672±0.0081|0.6077±0.0068|0.6486±0.0288|0.5954±0.0068|
> ||C|0.4222±0.2690|0.4523±0.2803|0.3514±0.3013|0.3399±0.2682|
> ---
>
> ## Discussion
> Here, although the generative process is linear and variables $X_0$ to $X_4$ have strong effects in both A and B, the signs of their coefficients are reversed between populations. This reduces the effectiveness of LASSO, which tends to select features based on average effects across all data. As a result, vanilla LASSO does not achieve the best performance even in this linear setting.
>
> For budget=$5$, our method outperforms most baselines, and has a balanced performance across population. For budget=$10$, our method is comparable with the best performing baselines.
>
> We select those features $i$ whose corresponding $\alpha_{i}$ values are lowest.
> Our method consistently selects a similar core ordered set of features across (3) different seeds (in decreasing order of importance) ``[0, 7, 1, 5, 8, 6, 9, 2, 10, 3], [0, 7, 1, 8, 5, 9, 6, 2, 10, 3],[0, 7, 5, 1, 8, 6, 9, 2, 10, 3]``. In contrast, baseline methods, especially non-DRO versions, show significant variability, often selecting irrelevant noisy features (e.g., 13, 14) and failing to consistently identify the true signal variables.

---

> > ### Comment · Reviewer_Guhq · 2025-08-03
> >
> > I would like to thank the authors for addressing my comments. However, one crucial issue remains unresolved, namely, the selection of appropriate baseline methods (as raised in Q2).
> >
> > Classical feature selection methods are generally divided into three categories:
> >
> > 1. Filter methods – which are not relevant in this context.
> > 2. Wrapper methods – which involve searching over feature subsets by repeatedly training a model.
> > 3. Embedded methods – which incorporate feature selection directly into the model training process by enforcing sparsity.
> >
> > Your proposed method clearly falls into the third category. The original baseline you used, LASSO, is an embedded method as well. However, the newly added baselines are wrapper methods, which are not a fair comparison.
> >
> > To convincingly demonstrate the superiority of your approach, it is essential to compare it against other embedded methods, particularly those designed for feature selection in heterogeneous or diverse populations. For example:
> >
> > Sample-wise feature selection:
> > Yang, Junchen, Ofir Lindenbaum, and Yuval Kluger. *"Locally sparse neural networks for tabular biomedical data."* International Conference on Machine Learning, PMLR, 2022.
> >
> > Cluster-level feature selection:
> > Svirsky, Jonathan, and Ofir Lindenbaum. *"Interpretable Deep Clustering for Tabular Data."* International Conference on Machine Learning, PMLR, 2024.
> >
> > These (and potentially other) relevant works should be discussed in the Related Work section and included as baselines in the Results section.

---

> ### Author Response · Authors · 2025-08-04
>
> We thank the reviewer for their response. We recognize the importance of comparing our method with other embedded methods. To this end, we add a new 'dro-embedded-mlp' baseline that uses an MLP with a learnable feature mask trained via DRO. We include the tabulated results at the end of this comment.
>
> With respect to the papers shared by the reviewer, we acknowledge these as new works in the space of feature selection, and will include them in our discussion on related works. However, these methods address different problem settings from our Group-DRO setting, and thus cannot be directly translated to our task:
> *Yang et al. (LSPIN)* and *Svirsky & Lindenbaum (IDC)* select different features for individual samples and clusters respectively—while our method requires selecting a single global feature subset that works universally across all known population groups (e.g., health systems often adopt screeners universally for all patients instead of asking an entirely custom set of questions for each patient or subgroup).
>
> Both methods allow different features for different samples/clusters, which does not align with our constraint of selecting a fixed feature subset for universal collection across populations. These represent complementary approaches to feature selection rather than direct alternatives to our distributionally robust framework. The new baselines implemented by us adapt the approach of the first paper suggested by the reviewer. However, instead of a sample-specific mask, we learn a global mask over features for all samples to better suit our Group DRO setting.
>
> # Experiments with embedded baselines
>
> To the results shared in our rebuttal we now add two new baselines (Embedded MLP 1, Embedded MLP 2). Both use an MLP with a learnable feature mask trained via DRO. MLP 1 has a single hidden layer of size 100, while MLP 2 has two hidden layers of sizes 64 and 32. We train both models with the joint objective of MSE minimization (for the regression task), and L1 regularization (weighted by hyperparameter $\lambda=0.01$ ; we found this $\lambda$ value to work best out of ``[0.1, 0.01, 0.001]``). Our method outperforms these baselines as well, with lower variance.
>
> |Method|Population|MSE±Std (MLP) — Budget 5|MSE±Std (RF) — Budget 5|MSE±Std (MLP) — Budget 10|MSE±Std (RF) — Budget 10|
> |---|---|---|---|---|---|
> |Our Method|A|0.2251±0.0029|0.2486±0.0018|0.0342±0.0004|0.0535±0.0010|
> ||B|0.2888±0.0113|0.3194±0.0098|0.0207±0.0005|0.0804±0.0036|
> ||C|0.2787±0.0150|0.3039±0.0130|0.0039±0.0004|0.0339±0.0079|
> |(BL) DRO Lasso|A|0.5071±0.0129|0.5434±0.0191|0.0338±0.0009|0.0534±0.0009|
> ||B|0.2886±0.0086|0.3194±0.0026|0.0211±0.0001|0.0807±0.0041|
> ||C|0.2755±0.0150|0.3031±0.0113|0.0039±0.0004|0.0340±0.0078|
> |(BL) DRO XGB|A|0.5107±0.0113|0.5441±0.0188|0.0738±0.0370|0.0887±0.0327|
> ||B|0.2888±0.0081|0.3187±0.0030|0.0408±0.0198|0.0928±0.0145|
> ||C|0.2762±0.0151|0.3032±0.0112|0.0036±0.0003|0.0337±0.0076|
> |(BL) Lasso|A|1.0087±0.0153|1.0571±0.0107|0.2056±0.1714|0.2106±0.1625|
> ||B|0.5744±0.0051|0.6150±0.0052|0.1160±0.0921|0.1460±0.0763|
> ||C|0.1080±0.0107|0.1283±0.0102|0.0036±0.0003|0.0340±0.0076|
> |(BL) XGB|A|1.0136±0.0076|1.0623±0.0039|0.4051±0.2802|0.3905±0.2579|
> ||B|0.5702±0.0008|0.6170±0.0076|0.2279±0.1586|0.2373±0.1307|
> ||C|0.1314±0.0291|0.1538±0.0352|0.0037±0.0002|0.0338±0.0077|
> |(BL) DRO Embedded MLP 1|A|0.7355±0.2702|0.7709±0.2636|0.5336±0.3503|0.5011±0.3107|
> ||B|0.5599±0.1363|0.6043±0.1310|0.4170±0.2951|0.4009±0.2521|
> ||C|0.2805±0.1630|0.3077±0.1728|0.2801±0.2117|0.2649±0.1923|
> |(BL) DRO Embedded MLP 2|A|0.7126±0.1538|0.7499±0.1467|0.0458±0.0735|0.0709±0.0640|
> ||B|0.8569±0.0729|0.8877±0.0748|0.5404±0.0494|0.5259±0.0436|
> ||C|0.4743±0.1496|0.5106±0.1601|0.5031±0.1682|0.4574±0.1339|

---

> > ### Comment · Reviewer_Guhq · 2025-08-04
> >
> > I thank the authors for the thorough clarifications and the additional simulations. You have fully addressed all of my comments.  Together with your responses to the other reviewers’ remarks, you resolved every concern raised during the review process. In light of these revisions, I am updating my score to 4. I wish you the best of luck with your submission!

---

### Official Review · Reviewer_ggn5 · 2025-07-03

**Clarity:** 4
**Significance:** 4
**Originality:** 4
**Rating:** 5
**Confidence:** 2

**Summary:**

This paper addresses the problem of selecting features such that models trained on these features are robust to distributional shifts. The authors formulate this problem as a continuous relaxation of the traditional variable selection using a noising mechanism. By optimziing over the variance of a Bayes-optimal predictor, they develop a model agnostic feature selectiom framework that achieves robust performance across populations. Experiments on synthetic and real-worlds datasets are conducted.

**Questions:**

See the weakness.

**Ethical Concerns:**

["NO or VERY MINOR ethics concerns only"]

**Final Justification:**

The author's response has addressed my concern. So I keep my positive score.

**Limitations:**

Yes

**Quality:**

3

**Strengths And Weaknesses:**

**Strength**

**1.**  This paper introduces an important problem of distributional robust feature selection, which lies at the intersaction of feature selection and domain generalization.

**2.** The continuous relaxation of feature selection is novel. It efficiently create a differetiable measure of feature utility and premit tracable optimization.

**3.** The writting of this paper is very clear and easy to follow.

**4.** The authos conduct extension experiments for validation

**Weakness**

**1.** The proposed method lacks theoretical optimality guarantee.

---

> ### Author Rebuttal · Authors · 2025-07-30
>
> We thank reviewer ggn5 for their feedback.
>
> We have conducted additional synthetic experiments and include three of them in this rebuttal.
> Our new experiments include:
> - Additional downstream model for evaluating performance: we now include an MLP (single hidden layer of size 100), along with the existing random forest (RF), as a downstream task model.
> - New baseline methods: we've added Forward Selection and Backward Elimination, and their DRO variants
> - Additional data generation settings to explore different covariate-outcome functional relationships.
>
> # Experiments
> For all experiments, we set dimension $d=50$, and use a 40-60 train-(test+val) split for each population
> # Expt 1: Linear generative process
> We use a dataset of size 36000 with the following composition:
> - A (40%): $$ Y = 8X_0 + 6X_1 - 4X_2 + 3X_3 + 2X_4 + \epsilon $$
> - B (35%): $$ Y = -8X_0 - 6X_1 + 4X_2 - 3X_3 - 2X_4 + 8X_5 + 6X_6 + \epsilon $$
> - C (25%): $$ Y = 10X_7 + 8X_8 + 6X_9 - 5X_{10} + \epsilon $$
> $$\text{Noise:} \quad \epsilon \sim \mathcal{N}(0,0.1^2)$$
>
> |Method|Population|MSE±Std (MLP) — Budget 5|MSE±Std (RF) — Budget 5|MSE±Std (MLP) — Budget 10|MSE±Std (RF) — Budget 10|
> |---|---|---|---|---|---|
> |Our Method| A|0.2227±0.0094| 0.2420±0.0161|0.0241±0.0178| 0.0450±0.0129|
> ||B|0.5805±0.0375| 0.6158±0.0448|0.0150±0.0096| 0.0809±0.0035|
> ||C|0.1208±0.0034| 0.1388±0.0011|0.0480±0.0761| 0.0697±0.0605|
> |(BL) DRO Lasso|A|0.4155±0.1769| 0.4418±0.1866|0.0345±0.0004| 0.0511±0.0029|
> ||B|0.2377±0.0909| 0.2677±0.0972|0.0206±0.0006| 0.0835±0.0067|
> ||C|0.3642±0.1350| 0.3849±0.1300|0.0040±0.0007| 0.0354±0.0012|
> |(BL) DRO XGB|A|0.6827±0.3136| 0.7120±0.3138|0.1041±0.0645| 0.1139±0.0597|
> ||B|0.3857±0.1461| 0.4209±0.1495|0.0613±0.0416| 0.1098±0.0326|
> ||C|0.2259±0.0928| 0.2479±0.0926|0.0040±0.0005| 0.0350±0.0010|
> |(BL) Lasso| A|1.0181±0.0526| 1.0429±0.0387|0.4126±0.5303| 0.3859±0.4760|
> ||B|0.5880±0.0323| 0.6264±0.0284|0.2327±0.2912| 0.2504±0.2401|
> ||C|0.1351±0.0252| 0.1550±0.0267|0.0037±0.0001| 0.0354±0.0009|
> |(BL) XGB| A|1.0211±0.0474| 1.0421±0.0400|1.1132±0.0439| 1.0216±0.0507|
> ||B|0.5867±0.0319| 0.6258±0.0282|0.6607±0.0314| 0.6135±0.0252|
> ||C|0.1356±0.0261| 0.1551±0.0267|0.0036±0.0004| 0.0347±0.0012|
> |(BL) DRO Forward Selection|A|0.3735±0.1506|0.3996±0.1636|0.0341±0.0006|0.0514±0.0029|
> ||B|0.3729±0.0553|0.3995±0.0553|0.0209±0.0012|0.0833±0.0066|
> ||C|0.4735±0.0805|0.5124±0.0883|0.0038±0.0001|0.0354±0.0012|
> |(BL) DRO Backward Elimination| A|0.0320±0.0012|0.0474±0.0024|0.0346±0.0005|0.0512±0.0030|
> ||B|0.1757±0.0093|0.2079±0.0149|0.0208±0.0008|0.0834±0.0068|
> ||C|1.0008±0.0669|1.0355±0.0635|0.0038±0.0003|0.0354±0.0012|
> |(BL) Forward Selection|A|1.0132±0.0372|1.0408±0.0435|1.1209±0.0483|1.0224±0.0403|
> ||B|0.5879±0.0331|0.6276±0.0384|0.6509±0.0360|0.6109±0.0362|
> ||C|0.9808±0.1161|1.0040±0.0989|1.1410±0.1413|0.9882±0.1095|
> |(BL) Backward Elimination|A|1.0095±0.0393|1.0463±0.0447|1.1267±0.0415|1.0244±0.0405|
> ||B|0.5897±0.0341|0.6267±0.0300|0.6640±0.0208|0.6111±0.0364|
> ||C|0.9883±0.1042|1.0071±0.0912|1.1047±0.1156|0.9895±0.1071|
> ---
>
> ## Discussion
> Here, although the generative process is linear and variables $X_0$ to $X_4$ have strong effects in both A and B, the signs of their coefficients are reversed between populations. This reduces the effectiveness of LASSO, which tends to select features based on average effects across all data. As a result, vanilla LASSO does not achieve the best performance even in this linear setting. For budget=$5$, our method outperforms most baselines, and has a balanced performance across population. For budget=$10$, our method is comparable with the best performing baselines.
>
> When we set dimension=$15$, our method is at par with the best performing baselines for all budgets- pl see the experiment section of our response to reviewer Guhq for the results.
>
> ## Expt 2: Mixed Linear-Nonlinear with Heterogeneous Noise
> We use a dataset of size 44000 with the following **population structure**:
>
> - A (30%):$$Y=4X_0+3X_1+X_2^2+\epsilon_A$$
> - B (30%):$$Y=4X_0+3X_1+X_2^2+\epsilon_B$$
> - C (25%):$$Y=2X_0+3X_5 X_6+4\sin(2X_7)+\epsilon_C$$
> - D (15%):$$Y=3X_0+2X_1+\epsilon_D$$
>
> **Heterogeneous Noise**:
> - $\epsilon_A \sim \mathcal{N}(0, (0.05)^2)$ (reduced noise)
> - $\epsilon_B=\exp(0.5X_3+0.3X_4) \cdot \eta \cdot 0.1$ where, $\eta \sim \mathcal{N}(0,1)$ (heteroscedastic)
> - $\epsilon_C \sim \mathcal{N}(0, 0.1^2)$ (standard)
> - $\epsilon_D \sim t_3 \cdot 0.2$ (heavy-tailed, t-distribution with df=3)
>
> ### Results
> |Method|Population|MSE±Std (MLP) — Budget 5|MSE±Std (RF) — Budget 5|MSE±Std (MLP) — Budget 8|MSE±Std (RF) — Budget 8|
> |:---|:---|:---|:---|:---|:---|
> |Our Method|A|0.0024±0.0005|0.0132±0.0032|0.0054±0.0007|0.0150±0.0033|
> ||B|0.0038±0.0010|0.0190±0.0016|0.0062±0.0010|0.0205±0.0016|
> ||C|0.8184±0.0750|0.8548±0.0918|0.0123±0.0080|0.2460±0.0483|
> ||D|0.0101±0.0003|0.0129±0.0007|0.0114±0.0015|0.0130±0.0008|
> |(BL) DRO Lasso|A|0.4107±0.0218|0.4333±0.0208|0.4115±0.0554|0.4020±0.0560|
> ||B|0.4199±0.0138|0.4499±0.0190|0.4172±0.0533|0.4230±0.0511|
> ||C|0.2778±0.2397|0.3782±0.1259|0.3114±0.2644|0.3816±0.1018|
> ||D|0.3225±0.0252|0.3408±0.0145|0.3569±0.0259|0.3375±0.0089|
> |(BL) DRO XGB|A|0.0026±0.0002|0.0133±0.0030|0.0054±0.0014|0.0150±0.0034|
> ||B|0.0037±0.0009|0.0189±0.0015|0.0064±0.0009|0.0206±0.0017|
> ||C|0.4312±0.0635|0.4690±0.0652|0.0226±0.0005|0.2460±0.0479|
> ||D|0.0102±0.0004|0.0127±0.0010|0.0114±0.0012|0.0132±0.0010|
> |(BL) Lasso|A|0.0510±0.0417|0.0617±0.0447|0.0517±0.0417|0.0616±0.0433|
> ||B|0.0547±0.0434|0.0653±0.0402|0.0562±0.0443|0.0657±0.0390|
> ||C|0.4310±0.0636|0.4611±0.0745|0.4654±0.0573|0.4581±0.0676|
> ||D|0.0100±0.0003|0.0126±0.0011|0.0114±0.0014|0.0130±0.0012|
> |(BL) XGB|A|0.0027±0.0007|0.0132±0.0031|0.0051±0.0013|0.0151±0.0033|
> ||B|0.0041±0.0012|0.0188±0.0015|0.0067±0.0012|0.0206±0.0015|
> ||C|0.4285±0.0622|0.4637±0.0613|0.0200±0.0039|0.2471±0.0475|
> ||D|0.0100±0.0002|0.0129±0.0010|0.0117±0.0017|0.0131±0.0007|
> |(BL) DRO Forward Selection|A|0.3881±0.0598|0.4151±0.0607|0.4088±0.0707|0.4066±0.0570|
> ||B|0.3989±0.0510|0.4239±0.0535|0.4208±0.0555|0.4186±0.0489|
> ||C|0.2765±0.2391|0.3717±0.1236|0.2980±0.2530|0.3746±0.1020|
> ||D|0.3253±0.0192|0.3467±0.0118|0.3522±0.0265|0.3391±0.0083|
> |(BL) DRO Backward Elimination|A|0.3889±0.0516|0.4152±0.0548|0.4076±0.0713|0.4060±0.0579|
> ||B|0.3996±0.0496|0.4309±0.0482|0.4188±0.0563|0.4191±0.0494|
> ||C|0.2771±0.2395|0.3685±0.1210|0.2930±0.2507|0.3756±0.1046|
> ||D|0.3242±0.0151|0.3460±0.0113|0.3519±0.0309|0.3392±0.0062|
> |(BL) Forward Selection|A|0.0506±0.0421|0.0607±0.0441|0.0531±0.0422|0.0613±0.0431|
> ||B|0.0545±0.0440|0.0654±0.0403|0.0569±0.0436|0.0654±0.0389|
> ||C|0.8114±0.0705|0.8523±0.0810|0.8811±0.0775|0.8445±0.0760|
> ||D|0.0099±0.0007|0.0124±0.0010|0.0115±0.0013|0.0130±0.0012|
> |(BL) Backward Elimination|A|0.0504±0.0417|0.0609±0.0442|0.0537±0.0434|0.0614±0.0431|
> ||B|0.0551±0.0438|0.0654±0.0408|0.0577±0.0449|0.0655±0.0388|
> ||C|0.8173±0.0702|0.8558±0.0753|0.8585±0.0741|0.8437±0.0742|
> ||D|0.0100±0.0003|0.0127±0.0013|0.0113±0.0020|0.0129±0.0012|
>
> ## Discussion
> For budget=$10$, our method is comparable with the best performing baseline (XGB). For the smaller budget, lasso and DRO have comparable performance, followed by our method. Forward selection and Backward Elimination consistently underperform other methods.
>
> ## Expt 3: Sparse Linear
> We use a dataset of size 36000 with the following composition:
>
> - A (35%):$$Y=5X_0+4X_{15}+3X_{30}+\epsilon$$
> - B (35%):$$Y=6X_5+5X_{20}+4X_{35}+\epsilon$$
> - C (30%):$$Y=7X_{10}+6X_{25}+5X_{40}+4X_{45}+\epsilon$$
>
> **Noise**:
> - $\epsilon\sim\mathcal{N}(0, 0.1^2)$ (base noise)
> - Correlated noise features: $X_{i+1}=0.3X_i+0.7\eta$ where $\eta \sim \mathcal{N}(0,1)$
>
> ### Results
> |Method|Population|MSE±Std (MLP) — Budget 5|MSE±Std (RF) — Budget 5|MSE±Std (MLP) — Budget 10|MSE±Std (RF) — Budget 10|
> |:---|:---|:---|:---|:---|:---|
> |Our Method|A|0.1186±0.0055|0.1299±0.0064|0.1274±0.0054|0.1277±0.0047|
> ||B|0.2099±0.0061|0.2290±0.0086|0.0034±0.0001|0.0093±0.0014|
> ||C|0.6009±0.0356|0.6453±0.0318|0.0036±0.0002|0.0282±0.0021|
> |(BL) DRO Lasso|A|0.1179±0.0053|0.1296±0.0066|0.0038±0.0003|0.0095±0.0003|
> ||B|0.2115±0.0070|0.2295±0.0088|0.0033±0.0004|0.0092±0.0013|
> ||C|0.6054±0.0438|0.6456±0.0310|0.0034±0.0001|0.0281±0.0021|
> |(BL) DRO XGB|A|0.1189±0.0057|0.1298±0.0066|0.0038±0.0000|0.0095±0.0004|
> ||B|0.2118±0.0071|0.2286±0.0092|0.0035±0.0003|0.0092±0.0014|
> ||C|0.6073±0.0432|0.6447±0.0311|0.0037±0.0004|0.0280±0.0018|
> |(BL) Lasso|A|0.4729±0.0153|0.5066±0.0185|0.0039±0.0004|0.0095±0.0004|
> ||B|0.2115±0.0057|0.2309±0.0090|0.0036±0.0004|0.0092±0.0013|
> ||C|0.3202±0.0029|0.3485±0.0106|0.0037±0.0001|0.0281±0.0018|
> |(BL) XGB|A|1.0188±0.0235|1.0651±0.0303|0.0036±0.0003|0.0095±0.0004|
> ||B|0.1414±0.1214|0.1564±0.1285|0.0034±0.0002|0.0091±0.0013|
> ||C|0.1915±0.1125|0.2121±0.1196|0.0037±0.0004|0.0280±0.0019|
> |(BL) DRO Forward Selection|A|0.5335±0.1147|0.5619±0.1252|0.0470±0.0746|0.0488±0.0687|
> ||B|0.5032±0.0173|0.5417±0.0190|0.0826±0.1371|0.0830±0.1267|
> ||C|0.6510±0.1143|0.6876±0.1048|0.2283±0.0848|0.2277±0.0756|
> |(BL) DRO Backward Elimination|A|0.1182±0.0049|0.1297±0.0065|0.0039±0.0005|0.0095±0.0003|
> ||B|0.2118±0.0074|0.2293±0.0087|0.0035±0.0004|0.0092±0.0014|
> ||C|0.6059±0.0430|0.6457±0.0309|0.0034±0.0003|0.0279±0.0019|
> |(BL) Forward Selection|A|0.4928±0.1395|0.5228±0.1508|0.3298±0.3016|0.3049±0.2713|
> ||B|0.0013±0.0001|0.0075±0.0012|0.0036±0.0004|0.0091±0.0013|
> ||C|0.7542±0.2417|0.7885±0.2349|0.5878±0.1616|0.5532±0.1395|
> |(BL) Backward Elimination|A|0.4056±0.2588|0.4297±0.2712|0.3312±0.3015|0.3055±0.2717|
> ||B|0.3015±0.1603|0.3233±0.1707|0.0035±0.0003|0.0091±0.0012|
> ||C|0.7559±0.2519|0.7844±0.2232|0.5951±0.1720|0.5529±0.1390|
>
> ## Discussion
> For budget=$5$, our method is comparable with the best performing baseline (XGB and DRO XGB). Despite the linear setting, lasso is not the best performing method. For budget=$10$, lasso and XGB variants, along with DRO Backward Elimination perform best. Our method succeeds in selecting all relevant variables except $X_{30}$, resulting in relatively worse performance on A with budget=$10$.

---

> > ### Comment · Reviewer_ggn5 · 2025-08-03
> >
> > I thank the author for the detailed response. I will keep my positive score.

---

### Comment · Area_Chair_1ywk · 2025-08-07
**Noise injection**

Thank you for your rebuttal and participation in the discussion.

Upon reviewing the submission, I noted the following claim (lines 131 ff):

    “[…] we introduce an alternative continuous relaxation which incorporates a random component controlled by α; effectively, αᵢ will control the amount of noise added to the observation of Xᵢ. To our knowledge, this relaxation is new to the literature.”

However, the idea of using noise as a regularization mechanism has a long history, dating back at least to 1995 [1]. It has also been formalized in the Bayesian framework [2,3], and optimization over α (in the notation of your paper) has been explored for variable selection in the standard setting (by this, I mean without the distributionally robust treatment) [4], including experiments in discrimination with non-linear models such as MLPs.

Can you provide some feedback regarding this point?

[1] Bishop CM. Training with noise is equivalent to Tikhonov regularization. Neural computation. 1995 Jan;7(1):108-16.

[2] R.M. Neal. Bayesian Learning for Neural Networks, Springer-Verlag, New York, 1996.

[3] M.E. Tipping. Sparse Bayesian learning and the relevance vector machine. Journal of Machine Learning Research, vol. 1, pp. 211–244, 2001.

[4] Grandvalet Y. Anisotropic noise injection for input variables relevance determination. IEEE Transactions on Neural Networks. 2000 Nov 30;11(6):1201-12.

---

> ### Author Response · Authors · 2025-08-08
>
> We thank the Area Chair for bringing to our attention the reference [4] (Grandvalet, 2000), and for the broader comment on literature on noise injection for regularization. We originally meant to claim that the use of noise was novel as a relaxation of 0-1 variable selection (as opposed to the broader use of noise as a form of regularization to improve predictive performance). However, we were not aware of [4] (Grandvalet, 2000)  and acknowledge this oversight in our literature review. We do wish to emphasize that all of the main technical developments (Sections 3.2-3.4) are still novel.
>
> We will revise the related work section accordingly. Specifically, we will replace the sentence on lines 131ff with the following, more detailed explanation of how our contributions relate to prior work and where they differ:
>
> > Previous work has proposed to use the injection of random noise to identify the most important variables for a predictive task in a single population [Grandvalet, 2000], as well as through Bayesian relevance estimation methods [Neal, 1996; Tipping, 2001]. However, tunable-noise-based variable selection has seen limited adoption since it was originally proposed, in contrast to the large body of work on noise as a form of regularization [Bishop, 1995].
> In this work, we revisit noise-based relaxations in the distributionally robust setting and show how the optimization problem can be reformulated in ways that create significant, previously-unrecognized advantages. In particular, our approach separates variable selection from fitting the predictive model, so that variable selection is agnostic with respect to the downstream model used for each different subpopulation. This also avoids the need to differentiate through the predictive models, as required in previous work (which may be impossible in the case of frequently-employed models like decision trees or random forests).
>
> Additionally, we wish to highlight that our contributions remain distinct from prior works in the following ways:
>
> 1. Distributionally robust setting: previous work does not discuss the DRO setting, the main motivation for our paper
> 2. Model agnostic framework: Our approach works with arbitrary downstream models (including non-differentiable ones like random forests) by targeting the performance of the Bayes-optimal predictor rather than specific trained models.
> 3. Training decoupled optimization: We provide a tractable, gradient based optimization framework directly over the task of feature selection, which can make use of a single set of pretrained models as input.

---

### Decision · Program_Chairs · 2025-09-17

**Decision:**

Accept (poster)

**Comment:**

This paper proposes a novel approach for feature selection in a distributionally robust context. The problem is framed as a continuous relaxation of hard variable selection, and relies on noise injection. The method decouples feature selection from downstream model training, and is thus applicable to non-differentiable models such as decision trees and random forests.

The reviewers concur that this is a relevant problem, at the intersection of feature selection and out-of-distribution generalization. The proposed method is technically sound. Empirical results on synthetic and real-world datasets demonstrate improvements over several baselines, in some cases by large margins. The availability of anonymized code and clear reporting of training details further strengthen the contribution. The experimental evaluation could consider more distribution shifts and more varied downstream models. Additional empirical evidence such as the one provided for the rebuttal will help support the claims. Also, an assessment of the robustness of the approach with respect to the Bayesian optimal predictor surrogate would be welcomed.

The consensus is that the paper offers novel and impactful contributions. Its formulation of distributionally robust feature selection is original, the proposed formalization is simple and the approach is generic and efficient. While further empirical validation would further strengthen the work, the novelty and potential impact of the paper justify acceptance at NeurIPS.